# Dataset Ownership Verification in Contrastive Pre-trained Models

**Yuechen Xie**[1], **Jie Song**[1]*, **Mengqi Xue**[2], **Haofei Zhang**[1], **Xingen Wang**[1,3], **Bingde Hu**[1,3],
**Genlang Chen**[4], **Mingli Song**[1,5,6]

[1]Zhejiang University, [2]Hangzhou City University, [3]Bangsheng Technology Co., Ltd
[4]NingboTech University, [5]State Key Laboratory of Blockchain and Security, Zhejiang University
[6]Hangzhou High-Tech Zone (Binjiang) Institute of Blockchain and Data Security

{xyuechen, sjie, haofeizhang, newroot, tonyhu, cgl, brooksong}@zju.edu.cn, mqxue@hzcu.edu.cn

## Abstract

High-quality open-source datasets, which necessitate substantial efforts for curation, has become the primary catalyst for the swift progress of deep learning. Concurrently, protecting these datasets is paramount for the well-being of the data owner. Dataset ownership verification emerges as a crucial method in this domain, but existing approaches are often limited to supervised models and cannot be directly extended to increasingly popular unsupervised pre-trained models. In this work, we propose the first dataset ownership verification method tailored specifically for self-supervised pre-trained models by contrastive learning. Its primary objective is to ascertain whether a suspicious black-box backbone has been pre-trained on a specific unlabeled dataset, aiding dataset owners in upholding their rights. The proposed approach is motivated by our empirical insights that when models are trained with the target dataset, the unary and binary instance relationships within the embedding space exhibit significant variations compared to models trained without the target dataset. We validate the efficacy of this approach across multiple contrastive pre-trained models including SimCLR, BYOL, SimSiam, MOCO v3, and DINO. The results demonstrate that our method rejects the null hypothesis with a $p$-value markedly below $0.05$, surpassing all previous methodologies. Our code is available at `https://github.com/xieyc99/DOV4CL`.

## 1 Introduction

The success of deep learning is greatly dependent on the the availability of high-quality open-source datasets, which empower researchers and developers to train and test their models and algorithms. Presently, the majority of public datasets Deng et al. (2009); Krizhevsky et al. (2009); Netzer et al. (2011) are designated exclusively for academic purposes, with commercial use prohibited without explicit permission. Therefore, preventing the stealing of public datasets holds significant importance for the benefit of the data owners.

Numerous traditional techniques exist for data security, including encryption Boneh & Franklin (2001); Khamitkar, differential privacy Dwork (2006); Abadi et al. (2016), and digital watermarking Cox et al. (2002); Podilchuk & Delp (2001); Kadian et al. (2021). However, these methods fall short in protecting the copyrights of open-source datasets, as they either impede dataset accessibility or necessitate the knowledge of the training process of potentially suspicious models. Recently, dataset ownership verification (DOV) Guo et al. (2023); Li et al. (2022; 2023b) emerges as a novel defense measure to deter dataset theft. It allows defenders, *i.e.,* dataset owners, to demonstrate whether suspects have infringed upon their rights by ascertaining whether a suspicious black-box backbone has been pre-trained on their datasets. However, as most existing DOV techniques are designed solely for supervised models where verification relies on distances between data points and decision boundaries Li et al. (2018); Karimi et al. (2019); Karimi & Tang (2020), they are not directly

---

*Corresponding author

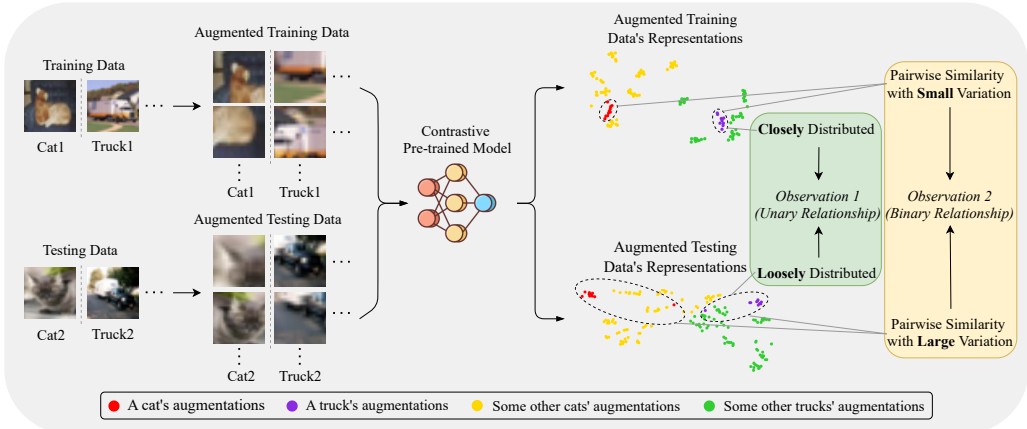

Figure 1: The overview of the two key observations. The representations are visualized using t-SNE. The encoder is a ResNet18 pre-trained on CIFAR10 with BYOL Grill et al. (2020).

applicable to recently increasing popular self-supervised pre-trained models Chen et al. (2020); Chen & He (2021); Chen et al. (2021a) due to the absence of the well-defined decision boundaries.

In this work, we present, to the best of our knowledge, the first DOV method for contrastive pre-trained models. It aids defenders in validating whether suspicious models have been illicitly pre-trained on their public datasets. Given a third-party suspicious model that might be pre-trained on the protected dataset without authorization, we focus on the black-box setting where defenders have no information about other training configurations (*e.g.*, loss function and model architecture) of the model and can only access model via Encoder as a Service (EaaS) Sha et al. (2023); Liu et al. (2022). It means defenders can only retrieve feature vectors via model API. The proposed approach is formulated upon two key observations, as shown in Figure 1. (1) *Unary relationship*: encoders pre-trained through contrastive learning generate remarkably more similar representations for augmentations of the same seen samples at the training phase than the unseen samples. (2) *Binary relationship*: the pairwise similarity between the seen samples doesn't significant change after data augmentations.

We define the differences in unary and binary relationships between seen and unseen samples as the *contrastive relationship gap* of the suspicious model. Defenders can endeavor to activate this gap in the suspicious encoder by employing their own public datasets, in order to ascertain whether the suspect's encoder was pre-trained on their data. More specifically, as illustrated in Figure 2, the proposed DOV technique comprises three steps: (1) pre-training a *shadow* encoder devoid of the public dataset of the defender; (2) utilizing multi-scale augmentation to compute the contrastive relationship gaps of the suspect encoder and the shadow encoder; (3) conducting hypothesis testing on the contrastive relationship gaps of the two encoders to determine whether the suspect encoder has been pre-trained on the defender's public dataset.

In summary, the principal contributions of this paper are threefold: (1) we discern that when models are trained with the target dataset, the unary and binary instance relationships within the embedding space demonstrate noteworthy disparities in comparison to models trained without the target dataset; (2) we introduce the concept of the contrastive relationship gap, which, to the best of our knowledge, represents the first DOV technique for contrastive pre-trained models; (3) comprehensive experiments showcase that our approach refutes the null hypothesis with a $p$-value significantly below $0.05$, surpassing all preceding studies.

## 2 RELATED WORK

**Data Protection.** Dataset ownership verification is an emerging field in data security. Typically, it involves embedding watermarks into the original dataset (Guo et al., 2023; Li et al., 2022; 2023b; Zhang et al., 2024; Xie et al., 2025). Models trained on the watermarked dataset will incorporate a pre-designed backdoor, allowing defenders to verify data ownership simply by triggering the model's backdoor. However, current DOV methods primarily target supervised models and require altering the original dataset's distribution to inject watermarks, which makes it susceptible to various watermark

removal mechanisms (Chen et al., 2021b; Liu et al., 2021b; Sun et al., 2023; Kwon, 2021; Hayase et al., 2021). The proposed method demonstrates that, for contrastive learning, dataset ownership can be efficiently verified without modifying the original dataset.

Dataset inference Maini et al. (2021) is a state-of-the-art defense against model stealing (Sha et al., 2023; Sanyal et al., 2022; Shen et al., 2022). It does not require retraining the model or embedding watermarks within the dataset, which reduces the time cost significantly while preserving the original distribution of the data. The latest dataset inference method Dziedzic et al. (2022) has expanded its application to self-supervised learning. Although it's primarily aimed at encoder theft, it can also be directly used for dataset ownership verification. However, it necessitates inferring the entire training set to model the output features of all data from both the training and testing sets. It is prohibitively time-consuming for large datasets, such as ImageNet (Deng et al., 2009). In contrast, our method achieves accurate verification using only a small fraction of the dataset. For instance, on ImageNet, we use only 0.1% of the training set for verification.

Membership inference Shokri et al. (2017); Choquette-Choo et al. (2021); Carlini et al. (2022); Hu et al. (2022) aims to determine whether an input was part of the model's training dataset. EncoderMI Liu et al. (2021a) is a powerful method specifically designed for membership inference on encoders pre-trained via contrastive learning, which takes advantage of the overfitting tendencies of the image encoder. However, it directly trains the inferencer on high-dimensional representations that contain a large amount of redundant information, which leads to a heavy computational cost and increased training difficulty. In contrast, our method extracts the most critical information for verification from the representations, namely contrastive relationship gap, achieving effective verification without the need to train an inferencer.

Inspired by Proof of Learning (PoL) Jia et al. (2021); Fang et al. (2023); Zhao et al. (2024), Proof of Training Data (PoTD) Choi et al. (2024) is proposed to assist third-party auditor in validating which data were used to train models. It helps develop practical and robust tools for accountability in the large-scale development of artificial intelligence models. However, it entails substantial verification costs, as the model trainer (suspect) is required to disclose detailed training records to the verifier, including training data, training code, and intermediate checkpoints. In practical scenarios, if the models trained by the suspect possess significant commercial value, the suspect is seldom willing to comply with such disclosures. Our setup is more reflective of real-world scenarios, where the model is a black box, and the defender can only access its API.

**Contrastive Learning.** The central idea of contrastive learning Chen et al. (2020); Chen & He (2021); Albelwi (2022); He et al. (2020) is to enable the encoder to produce similar feature vectors for a pair of augmentations derived from the same input image (positive samples), and distinct feature vectors for augmentations derived from different input images (negative samples). Classical approaches like SimCLR Chen et al. (2020), MoCo He et al. (2020), SwAV Caron et al. (2020), utilize both positive samples (for feature alignment) and negative samples (for feature uniformity). Surprisingly, researchers notice that contrastive learning can also work well by only aligning positive samples, such as BYOL Grill et al. (2020) and DINO Caron et al. (2021). We follow some literatures Albelwi (2022); Gao et al. (2022) to coin these methods as a special type of contrastive learning, or contrastive learning without negatives. We make no strict distinction between these concepts here due to the clear context in this work. Our method is designed to protect the unlabeled datasets used in contrastive learning, thereby securing and fostering healthy development in this field.

## 3 The Proposed Method

### 3.1 Problem Formulation

In this study, we focus on the dataset ownership verification task in black-box scenarios. The problem involves two key player: the *defender* and the *suspect*. The defender, assuming the role of the dataset provider, endeavors to ascertain whether the suspect model, $\mathcal{M}_{sus}$, has been unlawfully trained on his public dataset $\mathcal{D}_{pub}$. $\mathcal{M}_{sus}$ can be classified into four scenarios based on its training datasets: ① $\mathcal{M}_{sus}$ is exclusively trained on the public dataset $\mathcal{D}_{pub}$ of the defender, indicating the occurrence of dataset misappropriation; ② $\mathcal{M}_{sus}$ is trained on a dataset that encompasses the designated public dataset $\mathcal{D}_{pub}$ along with an additional dataset $\mathcal{D}_{alt}$, signifying dataset misappropriation, albeit posing a more challenging DOV task than case ① due to the presence of $\mathcal{D}_{alt}$; ③ $\mathcal{M}_{sus}$ is trained on an

unrelated dataset $\mathcal{D}_{unre}$ outside the scope of the defender's public dataset, indicating the innocence of the suspect; ④ $\mathcal{M}_{sus}$ is trained on an alternative dataset $\mathcal{D}_{alt}$ that bears significant resemblance yet doesn't overlap with the public dataset $\mathcal{D}_{pub}$, suggesting the innocence of the suspect, albeit posing a more arduous DOV challenge than case ③. These four scenarios encompass nearly every conceivable real-world circumstance.

## 3.2 CONTRASTIVE RELATIONSHIP GAP

### 3.2.1 OBSERVATIONS AND DEFINITIONS

In contrastive learning, a pivotal training objective for encoders is to maximize the similarity between the representations of positive samples, which are different augmentations of the same training image. This training approach leverages the neural network's memory capacity, prompting the encoder to retain the features of the training data. As a result, we derive the following two significant insights:

**Observation 1 (Unary Relationship)** *Contrastive pre-trained encoders can produce more alike representations for the same seen samples' augmentations during pre-training than unseen samples.*

**Observation 2 (Binary Relationship)** *The pairwise similarity between the seen samples' representations hardly change after augmentations, unlike with unseen samples during pre-training.*

We characterize the disparity between familiar and unfamiliar data encountered during the training phase as the encoder's *contrastive relationship gap*, a metric that can aid defenders in discerning whether the queried encoder has been pre-trained on their dataset. The precise definition is as follows:

**Definition 1 (Contrastive Relationship Gap)** *Given a contrastive pre-trained encoder $\mathcal{M}$ and a dataset $\mathcal{D}$, the contrastive relationship gap of $\mathcal{M}$ is defined as:*

$$d\big(\mathcal{D}, \hat{\mathcal{D}}, \mathcal{M}, T\big) = \left\{ s_i - \hat{s}_i \Big| i \in \big[1, |\mathcal{S}|\big], s_i \in \mathcal{S}\big(\mathcal{D}, \mathcal{M}, T\big), \hat{s}_i \in \mathcal{S}\big(\hat{\mathcal{D}}, \mathcal{M}, T\big) \right\} \qquad (1)$$

*where $\hat{\mathcal{D}}$ is a dataset that $\mathcal{M}$ has not been pre-trained on. $T(\cdot)$ denotes an augmentation function. $\mathcal{S}(\cdot, \cdot, \cdot)$ is a similarity set. $|\mathcal{S}|$ is the total number of samples in $\mathcal{S}(\mathcal{D}, \mathcal{M}, T)$.*

A larger mean of contrastive relationship gap suggests that $\mathcal{M}$ is more likely to have been pre-trained on $\mathcal{D}$. According to Observation 1 and Observation 2, $\mathcal{S}$ consists of unary relationship similarity set $\mathcal{S}_U$ and binary relationship similarity set $\mathcal{S}_B$.

### 3.2.2 THE CALCULATION OF $\mathcal{S}_U$ AND $\mathcal{S}_B$

Random cropping is a commonly used data augmentation technique in contrastive learning Chen et al. (2020); Chen & He (2021); Chen et al. (2021a), which can enhance the model's generalization ability significantly. In this paper, we use multi-scale random cropping to capture both global and local features of objects. Specifically, we design the $T$ in Eq.(1) as a multi-scale augmentation function $T^{ms} = \{T^g, T^l\}$, hoping to activate the encoder's contrastive relationship gap from various dimensions. $T^g$ is the global augmentation function responsible for larger regions, while $T^l$ is the local augmentation function focusing on smaller regions. Through $T^{ms}$, we calculate $\mathcal{S}_U$ and $\mathcal{S}_B$ at multi-scale. Their definitions are as follows:

**Definition 2 (Unary Relationship Similarity Set)** *Given an encoder $\mathcal{M}$ and a dataset $\mathcal{D}$, the unary relationship similarity set is defined as:*

$$\mathcal{S}_U\big(\mathcal{D}, \mathcal{M}, T^{ms}\big) = \{S_U^{gg}, S_U^{ll}, S_U^{gl}\} \qquad (2)$$

*where $S_U^{gg}$, $S_U^{ll}$, and $S_U^{gl}$ respectively denote the unary relationship similarity between global and global views, local and local views, and global and local views.*

The specific formulas are as follows:

$$S_U^{gg} = \frac{2}{|\mathcal{D}|M(M-1)} \sum_{i=1}^{|\mathcal{D}|} \sum_{m=1}^{M} \sum_{n=m+1}^{M} sim\Big(\mathcal{M}\big(T_m^g(x_i)\big), \mathcal{M}\big(T_n^g(x_i)\big)\Big) \qquad (3)$$

$$S_U^{ll} = \frac{2}{|\mathcal{D}|N(N-1)} \sum_{i=1}^{|\mathcal{D}|} \sum_{m=1}^{N} \sum_{n=m+1}^{N} sim\Big(\mathcal{M}\big(T_m^l(x_i)\big), \mathcal{M}\big(T_n^l(x_i)\big)\Big) \tag{4}$$

$$S_U^{gl} = \frac{1}{|\mathcal{D}|MN} \sum_{i=1}^{|\mathcal{D}|} \sum_{m=1}^{M} \sum_{n=1}^{N} sim\Big(\mathcal{M}\big(T_m^g(x_i)\big), \mathcal{M}\big(T_n^l(x_i)\big)\Big) \tag{5}$$

where $x_i \in \mathcal{D}$, and $|\mathcal{D}|$ is the total number of samples in dataset $\mathcal{D}$. $M$ and $N$ are the execution number for $T^g$ and $T^l$, respectively. $T_m^g(x_i)$ denotes the $m$-th augmentation of $x_i$ by $T^g$, similarly for $T_n^g(x_i), T_m^l(x_i), T_n^l(x_i)$. $sim(\cdot, \cdot)$ represents the cosine similarity function.

**Definition 3 (Binary Relationship Similarity Set)** *Similar to $\mathcal{S}_U$, given an encoder $\mathcal{M}$ and a dataset $\mathcal{D}$, the binary relationship similarity set is defined as:*

$$\mathcal{S}_B\big(\mathcal{D}, \mathcal{M}, T^{ms}\big) = \{S_B^{gg}, S_B^{ll}, S_B^{gl}\} \tag{6}$$

*where $S_B^{gg}$, $S_B^{ll}$, and $S_B^{gl}$ is the binary relationship similarity between global and global views, local and local views, and global and local views respectively.*

We first introduce the binary relationship set $\mathcal{G}$. It includes the pairwise similarity between the augmented images' representations, denoted as:

$$\mathcal{G}(\mathcal{D}, \mathcal{M}, T) = \left\{ sim\Big(\mathcal{M}\big(T(x_i)\big), \mathcal{M}\big(T(x_j)\big)\Big) \Big| i \in [1, |\mathcal{D}|], j \in (i, |\mathcal{D}|] \right\} \tag{7}$$

where $sim(\cdot, \cdot)$ denotes the cosine similarity function, with $x_i, x_j \in \mathcal{D}$, $|\mathcal{D}|$ is the total number of samples in dataset $\mathcal{D}$, and $T(\cdot)$ represents the augmentation function. By substituting the augmentation functions $T^g$ and $T^l$ into Eq.(7), we obtain the binary relationship set $\mathcal{G}^g$ and $\mathcal{G}^l$ at respective scales. Below, we formally present the specific formulas for $S_B^{gg}, S_B^{ll}$, and $S_B^{gl}$:

$$S_B^{gg} = -\frac{2}{M(M-1)} \sum_{m=1}^{M} \sum_{n=m+1}^{M} f(\mathcal{G}_m^g, \mathcal{G}_n^g) \tag{8}$$

$$S_B^{ll} = -\frac{2}{N(N-1)} \sum_{m=1}^{N} \sum_{n=m+1}^{N} f(\mathcal{G}_m^l, \mathcal{G}_n^l) \tag{9}$$

$$S_B^{gl} = -\frac{1}{MN} \sum_{m=1}^{M} \sum_{n=1}^{N} f(\mathcal{G}_m^g, \mathcal{G}_n^l) \tag{10}$$

where $f(\cdot, \cdot)$ is a distance measurement function, which is implemented as the mean absolute error in this paper. $M$ and $N$ are the execution number for $T^g$ and $T^l$, respectively. $\mathcal{G}_m^g$ represents the $m$-th binary relationship set based on $T^g$, similarly for $\mathcal{G}_n^g, \mathcal{G}_m^l$ and $\mathcal{G}_n^l$.

Using unary relationship similarity set $\mathcal{S}_U$ and binary relationship similarity set $\mathcal{S}_B$, we can determine the contrastive relationship gap $d$ of the encoder $\mathcal{M}$ as follows:

$$d = \left\{ \sum_* (S_U^* - \hat{S}_U^*) \cdot I(S_U^* > \hat{S}_U^*), \sum_* (S_B^* - \hat{S}_B^*) \cdot I(S_B^* > \hat{S}_B^*) \right\} \tag{11}$$

where $* \in \{gg, ll, gl\}$, $S$ and $\hat{S}$ come from $\mathcal{S}(\mathcal{D}, \mathcal{M}, T)$ and $\mathcal{S}(\hat{\mathcal{D}}, \mathcal{M}, T)$ in Eq.(1), respectively. $I(\cdot)$ is the function returning $a$ if the input statement is true or returning 1 if the input statement is false. $a$ is a hyperparameter with a default value of 1. As $a$ increases, the contrastive relationship gap of encoder $\mathcal{M}$ between $\mathcal{D}$ and $\hat{\mathcal{D}}$ becomes larger.

### 3.2.3 THE COMPLETE PROCESS

We propose a method of dataset ownership verification by contrastive relationship gap. Figure 2 displays the entire process of our method, divided into three stages:

(1) pre-training a shadow encoder $\mathcal{M}_{sdw}$ on a shadow dataset $\mathcal{D}_{sdw}$ to compare with $\mathcal{M}_{sus}$;

(2) performing $K$ samplings on $\mathcal{D}_{pub}$ and $\mathcal{D}_{pvt}$ (a defender's private dataset which isn't publicly available, and $\mathcal{M}_{sus}$ has not been trained on it), that represent $\mathcal{D}$ and $\hat{\mathcal{D}}$ in Eq.(1), respectively. The sampling sizes are $k_{pub}$ and $k_{pvt}$ respectively, resulting in the subsets $\{\mathcal{D}_{pub}^1, \cdots, \mathcal{D}_{pub}^K\}$ and $\{\mathcal{D}_{pvt}^1, \cdots, \mathcal{D}_{pvt}^K\}$. Then using these subsets calculate the contrastive relationship gaps $d_{sus} = d_{sus}^1 \cup \cdots \cup d_{sus}^K$ and $d_{sdw} = d_{sdw}^1 \cup \cdots \cup d_{sdw}^K$ of $\mathcal{M}_{sus}$ and $\mathcal{M}_{sdw}$, respectively;

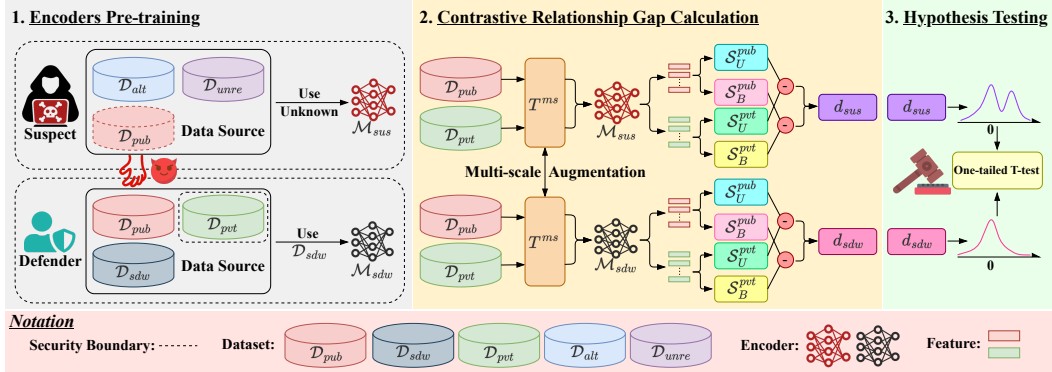

Figure 2: The overview of our method (best viewed under color conditions).

(3) one-tailed pair-wise T-test Hogg et al. (2013) is conducted on $d_{sus}$ and $d_{sdw}$. The null hypothesis, $H_0$, posits that the mean of $d_{sus}$ is less than or equal to that of $d_{sdw}$, while the alternative hypothesis, denoted as $H_1$, posits that the mean of $d_{sus}$ is greater than the mean of $d_{sdw}$. If the $p$-value $p$ is less than 0.05, we can reject the null hypothesis and conclude that $\mathcal{D}_{pub}$ has been stolen. On the other hand, if the null hypothesis can't be rejected, we think the suspect is innocent.

## 4 EXPERIMENTS

We evaluate our method using six visual datasets (CIFAR10 Krizhevsky et al. (2009), CIFAR100 Krizhevsky et al. (2009), SVHN Netzer et al. (2011), ImageNette Howard (2019), ImageWoof Howard (2019) and ImageNet Deng et al. (2009)) and five contrastive learning algorithms (SimCLR, BYOL, SimSiam, MOCO v3, and DINO). ImageNette and ImageWoof are two nonoverlapping subsets of ImageNet, each containing 10 classes.

The specific experimental setup is introduced in Section 4.1, results and analyses are presented in Section 4.2, the application of our method on the ImageNet pre-trained models are demonstrated in Section 4.3, ablation studies are conducted in Section 4.4 and Appendix A. Specifically, Appendix A.7 presents the ablation study of sampling size, the ablation study of global and local augmentation number is shown in Appendix A.8, and the ablation study of shadow dataset and hyperparameter $a$ are featured in Appendix A.9. The impact of shadow model's training hyperparameters is shown in Appendix A.5. The anti-interference capability of our method is conducted in Section 4.5, Appendix A.11 introduces the comparison with the method based on watermark. Appendix A.10 introduces the impact of early stopping. Appendix A.13 presents some visualization results of our method.

### 4.1 EXPERIMENTAL SETUP

For SimCLR, BYOL, SimSiam, and MoCo v3, we use VGG16 Simonyan & Zisserman (2014), and Resnet18 He et al. (2016) as encoder architectures. Additionally, we use ViT-T, ViT-S, and ViT-B Dosovitskiy et al. (2020) for DINO. For $\mathcal{M}_{sdw}$, we default to using ResNet18 and SimCLR as its encoder architecture and training algorithm.

To simulate $\mathcal{D}_{alt}$, a dataset similar to $\mathcal{D}_{pub}$ but without overlapping data (as described in Section 3.1), we randomly divide a dataset into two subsets of equal size representing $\mathcal{D}_{pub}$ and $\mathcal{D}_{alt}$, respectively. For $\mathcal{D}_{pvt}$, we set it as the testing set of the undivided dataset for convenience. Specific settings are as follows:

- **Experiment 1**: $\mathcal{D}_{pub}$ is random half of CIFAR10 training set and $\mathcal{D}_{alt}$ is the other half. $\mathcal{D}_{unre}$, $\mathcal{D}_{sdw}$ and $\mathcal{D}_{pvt}$ are SVHN, CIFAR100 and CIFAR10 testing set respectively.
- **Experiment 2**: $\mathcal{D}_{pub}$ is random half of ImageNette training set and $\mathcal{D}_{alt}$ is the other half. $\mathcal{D}_{unre}$, $\mathcal{D}_{sdw}$ and $\mathcal{D}_{pvt}$ are ImageWoof, SVHN and ImageNette testing set respectively.

The settings for the remaining parameters are provided in Appendix A.2. To simulate adversarial behavior, we pre-train $\mathcal{M}_{sus}$ using $\mathcal{D}_{pub}$, $\mathcal{D}_{pub} \cup \mathcal{D}_{alt}$, $\mathcal{D}_{unre}$, and $\mathcal{D}_{alt}$, respectively, which is corresponds to the four cases in Section 3.1.

Regarding evaluation metrics, in addition to using the $p$-value, we also use the sensitivity, specificity and AUROC. Sensitivity is the proportion of correctly predicted positive cases among all actual positive samples, and specificity is the proportion of correctly predicted negative cases among all actual negative samples. They reflect the ability to identify positive and negative samples, respectively.

When $\mathcal{M}_{sus}$ pre-trained on $\mathcal{D}_{pub}$ or $\mathcal{D}_{pub} \cup \mathcal{D}_{alt}$, which means the suspect is illegal, $p$ should be less than 0.05. When $\mathcal{M}_{sus}$ pre-trained on $\mathcal{D}_{alt}$ or $\mathcal{D}_{unre}$, which means the suspect is legal, $p$ should be greater than 0.05. We compare our method with two representative methods, as detailed below:

- **DI4SSL** Dziedzic et al. (2022): This is the most recent method for dataset inference targeting self-supervised encoders. It also applies to dataset ownership verification. The principle behind DI4SSL is that if the encoder is pre-trained on $\mathcal{D}_{pub}$, the representations it outputs will have a higher log-likelihood on the defender's training data than on testing data. Conversely, if the encoder is not pre-trained on $\mathcal{D}_{pub}$, this pattern will not be observed.

- **EncoderMI** Liu et al. (2021a): This is a classic method which designed for member inference on contrastive pre-trained models. The fundamental mechanism of EncoderMI is that the encoder produces similar representations for different augmentation of the training data. We have adapted this method to suit dataset ownership verification better. Specifically, we augment images from $\mathcal{D}_{pub}$ and $\mathcal{D}_{pvt}$ and input them into $\mathcal{M}_{sus}$ and $\mathcal{M}_{sdw}$. By comparing the distribution of the output representations' similarity, we can determine potential dataset stealing. If $\mathcal{M}_{sus}$ is pre-trained on $\mathcal{D}_{pub}$, the representations' similarity of $\mathcal{M}_{sus}$ will significantly exceed those of $\mathcal{M}_{sdw}$, vice versa.

## 4.2 EXPERIMENTAL RESULTS

Our approach is proven effective as illustrated in Figure 3 (refer to Appendix A.3 and A.4 for specific $p$-values), which display the experimental results of baselines and our method on CIFAR10 and ImageNette. Note that when $\mathcal{D}_{sus}$ is CIFAR10-1 (ImageNette-1) or CIFAR10 (ImageNette), $\mathcal{D}_{sus}$ includes $\mathcal{D}_{pub}$ ($\mathcal{D}_{pub}$ is CIFAR10-1 and ImageNette-1 in two cases respectively), which implies the suspect is illegal, and $p$ should be less than 0.05. However, when $\mathcal{D}_{sus}$ is CIFAR10-2 (ImageNette-2) or SVHN (ImageWoof), the suspect did not use $\mathcal{D}_{pub}$ and is legal, so $p$ should be greater than 0.05.

The two baselines struggle to accurately distinguish the legality of various scenarios. There are a large number of false positive or false negative samples in all cases. In contrast, our method consistently produces correct results in all cases. Unlike the baselines, which model high-dimensional representations containing a large amount of redundant information directly, our method refines the most valuable information from these representations.

This crucial information, contrastive relationship gap, is extracted based on the characteristics of contrastive learning. Therefore, our method is not constrained by the encoder architecture and training algorithm, achieving desirable outcomes in various scenarios. As shown in Table 1, we calculate sensitivity, specificity and AUROC based on the experimental results on CIFAR10 and ImageNette, which demonstrates the superiority of our method quantitatively.

Table 1: Sensitivity, specificity, and AUROC of three methods on CIFAR10 and ImageNette.

| Dataset | Method | Sensitivity | Specificity | AUROC |
|---------|--------|-------------|-------------|-------|
| CIFAR10 | DI4SSL | 0.2 | 1.0 | 0.6 |
| | EncoderMI | 0.8 | 0.2 | 0.5 |
| | Ours | **1.0** | **1.0** | **1.0** |
| ImageNette | DI4SSL | 0.3 | 1.0 | 0.775 |
| | EncoderMI | 0.15 | 0.9 | 0.5 |
| | Ours | **1.0** | **1.0** | **1.0** |

Sensitivity and specificity reflect the algorithm's ability to identify positive and negative samples.

## 4.3 THE APPLICATION OF OUR METHOD ON IMAGENET

To validate the efficacy of our method in real-world scenarios, we conduct dataset ownership verification on ImageNet, a large-scale visual dataset containing over 14 million images across 1000 classes, using ten pre-trained encoders. The architecture of these encoders includes CNN and ViT, and they are pre-trained using the six popular contrastive learning methods currently. Among these, the pre-trained model for DINO is obtained from the official repository[1], while the models for the

---

[1]https://github.com/facebookresearch/dino

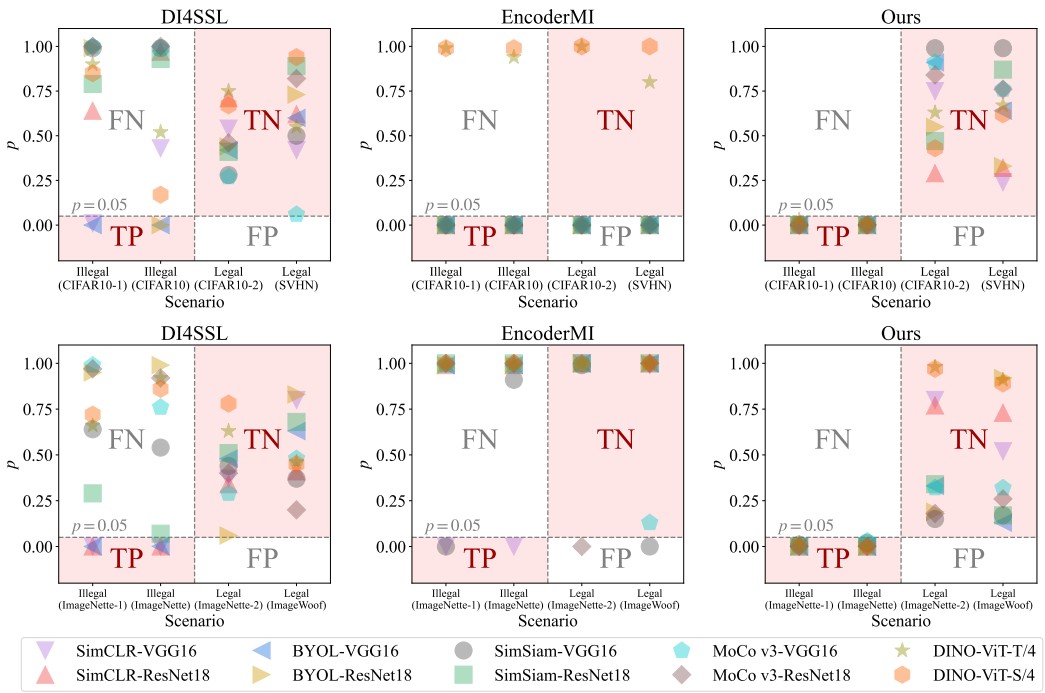

Figure 3: Experimental results of three methods on CIFAR10 (the first line) and ImageNette (the second line). Each value is an average of 3 trials. Each pattern represents a suspicious model trained using a specific architecture, contrastive learning method, and dataset. 'SimCLR-VGG16' represents VGG16 trained using SimCLR, and the rest follows similarly. 'CIFAR10-1' and 'CIFAR10-2' are the two non-overlapping random halves of CIFAR10 training set, similarly for 'ImageNette-1' and 'ImageNette-2'. $\mathcal{D}_{pub}$ is CIFAR10-1 and ImageNette-1 in two cases respectively. We consider illegal/legal behavior as positive/negative cases and classify each situation based on $p$-value. The datasets in parentheses on the x-axis are $\mathcal{D}_{sus}$.

other contrastive learning methods are sourced from MMSelfSup[2]. In our experiments, we designate $\mathcal{D}_{pvt}$ as the validation set of ImageNet and $\mathcal{D}_{sdw}$ as SVHN. The architecture and training algorithm of $\mathcal{M}_{sdw}$ are ResNet18 and SimCLR, respectively. Parameter settings are provided in Appendix A.2. As shown in Table 2, the experimental outcomes demonstrate that our method is well-suited for pre-trained models on ImageNet, even when using only 0.1% of ImageNet data for dataset ownership verification. Conversely, the performances of baselines are unsatisfactory.

## 4.4 ABLATION STUDIES

### 4.4.1 THE IMPACT OF MULTI-SCALE AUGMENTATION IN UNARY AND BINARY RELATIONSHIP

We use pre-trained models on ImageNet to verify the effectiveness and robustness of unary and binary relationship's multi-scale augmentations. Specifically, the models are ResNet50 and ViT-B/16 pre-trained by DINO. Both $\mathcal{D}_{pub}$ and $\mathcal{D}_{sus}$ are ImageNet. As shown in Table 3, The combined use of unary and binary relationship's multi-scale augmentations outperform other choices. This superiority is attributed to its attempts to activate the encoder's contrastive relationship gap from various angles, thereby endowing it with strong generalization capabilities to adapt to different encoders.

### 4.4.2 THE IMPACT OF SAMPLE NUMBER OF $\mathcal{D}_{pub}$ AND $\mathcal{D}_{alt}$

We study the impact of the sample number of $\mathcal{D}_{pub}$ and $\mathcal{D}_{alt}$ on our method. Specifically, we denote the proportion of $\mathcal{D}_{pub}$ in $\mathcal{D}_{pub} \cup \mathcal{D}_{alt}$ as $r$. And $\mathcal{D}_{pub} \cup \mathcal{D}_{alt}$ is always CIFAR10. For example, when $r = 0.1$, $\mathcal{D}_{pub}$ is 10% of the CIFAR10 training set randomly sampled, while $\mathcal{D}_{alt}$ consists of the remaining 90%. Similarly, when $r = 0.2$, $\mathcal{D}_{pub}$ is 20% of the CIFAR10 training set randomly sampled, and $\mathcal{D}_{alt}$ is the remaining 80%.

---

[2]https://mmselfsup.readthedocs.io/en/latest/model_zoo.html

Table 2: The results ($p$-values) of baselines and our method applied on ImageNet. '$\mathcal{D}_{sus}$' is the dataset used to pre-train $\mathcal{M}_{sus}$. Each value is an average of 3 trials. $\mathcal{D}_{sus}$ and $\mathcal{D}_{pub}$ are both ImageNet. Note that in this scenario, $\mathcal{D}_{sus}$ includes $\mathcal{D}_{pub}$, making the suspect's behavior illegal, and the $p$-values should be less than 0.05.

| Method | Model | DI4SSL | EncoderMI | Ours |
|---|---|---|---|---|
| SimCLR | | 0.15 | 1 | $10^{-3}$ |
| BYOL | ResNet50 | 0.91 | 1 | $10^{-3}$ |
| SimSiam | | 0.56 | 1 | $10^{-4}$ |
| SwAV | | 0.88 | 1 | $10^{-4}$ |
| | ResNet50 | 0.51 | 1 | $10^{-3}$ |
| MoCo v3 | ViT-S/16 | 0.99 | $10^{-159}$ | $10^{-4}$ |
| | ViT-B/16 | 0.99 | $10^{-158}$ | $10^{-4}$ |
| | ResNet50 | 0.99 | 1 | $10^{-4}$ |
| DINO | ViT-S/16 | 0.99 | 1 | $10^{-3}$ |
| | ViT-B/16 | 0.99 | 1 | $10^{-3}$ |

Table 3: The impact of multi-scale augmentation in unary and binary relationship. Both $\mathcal{D}_{pub}$ and $\mathcal{D}_{sus}$ are ImageNet. 'DINO-ResNet50' represents ResNet50 trained using DINO, with 'DINO-ViT-B/16' being similar. Note that the suspect is illegal in this case, and the $p$-values should be less than 0.05. **Bold** and underline respectively represent the best and second best results.

| Study Subject | $S_U^{gg}$ | $S_U^{gl}$ | $S_U^{ll}$ | $S_B^{gg}$ | $S_B^{gl}$ | $S_B^{ll}$ | DINO-ResNet50 | DINO-ViT-B/16 |
|---|---|---|---|---|---|---|---|---|
| Unary/Binary Relationship | ✓ | ✓ | ✓ | | | | 0.02 | 0.01 |
| | | | | ✓ | ✓ | ✓ | $3.1 \times 10^{-3}$ | $2.4 \times 10^{-3}$ |
| | ✓ | | | ✓ | | | 0.06 | 0.02 |
| | | ✓ | | | ✓ | | $\mathbf{3.3 \times 10^{-4}}$ | $2.5 \times 10^{-3}$ |
| Global/Local View | | | ✓ | | | ✓ | $3.9 \times 10^{-3}$ | $1.4 \times 10^{-3}$ |
| | ✓ | ✓ | | ✓ | ✓ | | $8.2 \times 10^{-4}$ | $2.5 \times 10^{-3}$ |
| | | ✓ | ✓ | | ✓ | ✓ | $5.0 \times 10^{-4}$ | $1.4 \times 10^{-3}$ |
| | ✓ | | ✓ | ✓ | | ✓ | $1.8 \times 10^{-3}$ | $\mathbf{1.0 \times 10^{-3}}$ |
| Ours | ✓ | ✓ | ✓ | ✓ | ✓ | ✓ | $4.2 \times 10^{-4}$ | $1.1 \times 10^{-3}$ |

Then we use ResNet18 pre-trained using SimCLR to obverse the performance changes of our method under different $r$ values. In Figure 4, each point represents the $p$-value (log-transformed) of the model trained on the corresponding dataset. It shows our method demonstrates good robustness to the sample number of $\mathcal{D}_{pub}$ and $\mathcal{D}_{alt}$.

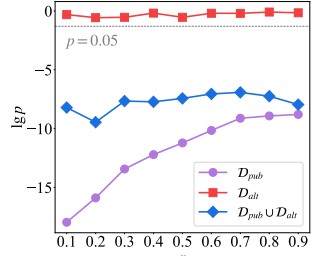

## 4.5 THE ANTI-INTERFERENCE CAPABILITY OF OUR METHOD

### 4.5.1 THE IMPACT OF PRIVACY TRAINING METHOD

Private training methods Abadi et al. (2016); Papernot et al. (2018) are typically used to protect private, non-open-source datasets. In our scenario, the suspect might employ private training methods to obscure their illegal activities and interfere with the defender's dataset ownership verification, even if it reduces the encoder's normal performance. Therefore, we chose the classic private training

Figure 4: The impact of the ratio of $\mathcal{D}_{pub}$ to $\mathcal{D}_{pub} \cup \mathcal{D}_{alt}$ on our method. Each point is the $p$-value (log-transformed) of the model trained on the corresponding dataset.

method DP-SGD Abadi et al. (2016) and conducted the following experiments. Specifically, we trained the suspicious encoder on ImageNette using DP-SGD or not. The $\epsilon$ for DP-SGD is 50, and the maximum norm for gradient clipping is 1.2. The results are shown in Table 4 and Table 5, indicating that our method remains effective in this more arduous scenario.

Table 4: Results ($p$-values) on SimCLR.

| Model | w/o DP-SGD | w/ DP-SGD |
|---|---|---|
| VGG16 | $10^{-27}$ | $10^{-14}$ |
| ResNet18 | $10^{-14}$ | $10^{-13}$ |

Table 5: Results ($p$-values) on SimSiam.

| Model | w/o DP-SGD | w/ DP-SGD |
|---|---|---|
| VGG16 | 0.01 | 0.01 |
| ResNet18 | $10^{-4}$ | $10^{-3}$ |

### 4.5.2 THE APPLICATION OF OUR METHOD ON FINE-TUNED ENCODERS

We also challenge the scenario where $\mathcal{M}_{sus}$ is applied to downstream tasks. Specifically, we train the entire classifier on CIFAR10 and CIFAR100 respectively, whose backbone is a ResNet50 pre-trained on ImageNet using SimCLR. Similarly, in the black-box environment, we can only use the predicted probability vectors of the input samples. The results are shown in Table 6 and Table 7. '$\mathcal{D}_{downstream}$' is the dataset of downstream tasks. 'Acc' represents the accuracy on downstream tasks. $\mathcal{D}_{sus}$ and $\mathcal{D}_{pub}$ are both ImageNet. Note that in this scenario, $\mathcal{D}_{sus}$ includes $\mathcal{D}_{pub}$, making the suspect's behavior illegal, and the $p$-values should be less than 0.05. Moreover, we set the hyperparameter $a$ to 5 to amplify the contrastive relationship gap. Excitingly, even after fine-tuning, we are still able to identify the suspect's theft. For details on fine-tuning, please refer to Appendix A.6.

Table 6: Fine-tuning Results on CIFAR-10.

| $\mathcal{D}_{downstream}$ | Epoch | $p(\downarrow)$ | Acc |
|---|---|---|---|
| | 50 | $10^{-4}$ | 0.87 |
| CIFAR10 | 100 | $10^{-3}$ | 0.88 |
| | 150 | $10^{-8}$ | 0.88 |
| | 200 | $10^{-4}$ | 0.89 |

Table 7: Fine-tuning Results on CIFAR-100.

| $\mathcal{D}_{downstream}$ | Epoch | $p(\downarrow)$ | Acc |
|---|---|---|---|
| | 50 | $10^{-6}$ | 0.44 |
| CIFAR100 | 100 | $10^{-4}$ | 0.50 |
| | 150 | $10^{-5}$ | 0.63 |
| | 200 | $10^{-3}$ | 0.66 |

### 4.6 THE TIME COST OF OUR METHOD

We calculated the time required for our method and DI4SSL to perform a single verification on ImageNet. The experiments were conducted using an NVIDIA GeForce RTX 4090. The encoder is a ResNet50 pre-trained on ImageNet using SimCLR. As shown in Table 8, the time consumption of our method is significantly less than that of DI4SSL.

Table 8: The time required for our method and DI4SSL to execute once on ImageNet.

| Method | Time Consumption | $p(\downarrow)$ |
|---|---|---|
| DI4SSL | 10014s | 1 |
| Ours | 293s | $10^{-3}$ |

This is because our method only requires inferring on a small subset of ImageNet (depending on $k_{pub}$, $k_{pvt}$, $M$ and $N$), whereas DI4SSL needs to infer the entire dataset. Additionally, our method was properly validated ($p < 0.05$), further demonstrating its superiority.

### 4.7 LIMITATIONS

Not all encoders are pre-trained using contrastive learning. Masked Image Modeling (MIM) Girdhar et al. (2023); He et al. (2022) is also a significant method for pre-training encoders. However, as shown in Appendix A.12, our method doesn't effectively apply to encoders pre-trained via MIM. This is because that the representations learned through MIM are harder to distinguish compared to those from contrastive learning Zhou et al. (2022), although MIM-based pre-training methods demonstrate superior performance in downstream tasks. This results in less pronounced unary and binary relational gaps in the representations. We plan to refine this aspect in our future work.

## 5 CONCLUSION

High-quality open-source datasets are essential for the rapid development of deep learning. We propose a method for verifying dataset ownership in contrastive learning to protect the legitimate right of dataset owners. Specifically, we propose the concept of contrastive relationship gap based on the unary and binary relationship of contrastive pre-trained models. The experiment proves that it can effectively verify dataset ownership. Promising future work includes (1) extending our method to other self-supervised learning approaches; (2) adapting our method to protect other types of data (e.g., text); (3) exploring other privacy risks associated with encoders.

## 6 ACKNOWLEDGEMENTS

This work was partially supported by the Pioneer R&D Program of Zhejiang (No.2024C01021), and Zhejiang Provincial Natural Science Foundation of China (LQ24F020020, LD24F020011).

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

APPENDIX

## A   THE DETAILS AND ADDITIONAL SUPPLEMENTS OF EXPERIMENTS

### A.1   DATASETS USED

**CIFAR10** Krizhevsky et al. (2009): The CIFAR10 dataset consists of 32x32 colored images with 10 classes. There are 50000 training images and 10000 test images.

**CIFAR100** Krizhevsky et al. (2009): The CIFAR100 dataset consists of 32x32 coloured images with 100 classes. There are 50000 training images and 10000 test images.

**SVHN** Netzer et al. (2011): The SVHN dataset contains 32x32 coloured images with 10 classes. There are roughly 73000 training images, 26000 test images and 530000 "extra" images.

**ImageNette** Howard (2019): ImageNette is a subset of 10 easily classified classes from Imagenet. It includes the following categories: tench, English springer, cassette player, chain saw, church, French horn, garbage truck, gas pump, golf ball and parachute. There are roughly 10000 training images and 4000 test images.

**ImageWoof** Howard (2019): ImageWoof is a subset of 10 classes from Imagenet that aren't so easy to classify. It includes the following categories: Australian terrier, Border terrier, Samoyed, Beagle, Shih-Tzu, English foxhound, Rhodesian ridgeback, Dingo, Golden retriever, Old English sheepdog. There are approximately 9000 training images and 4000 test images.

**ImageNet** Deng et al. (2009): Larger sized coloured images with 1000 classes. There are approximately 1 million training images and 50000 test images. As is commonly done, we resize all images to be of size 224x224.

### A.2   EXPERIMENTAL DETAILS

The ResNet18 trained on CIFAR10/CIFAR100 uses a convolutional kernel size of 3x3 with a stride of 1, instead of the default 7x7, and doesn't use a max pooling layer.

On CIFAR10/CIFAR100/SVHN, we pre-train the encoder for 800 epochs with a batch size of 512. On ImageNette/ImageWoof, the encoder with non-ViT-S/16 architecture is pre-trained for 800 epochs, while ViT-S/16 architecture is pre-trained for 2000 epochs with a batch size of 64. The initial learning rate for all pre-training sessions is set at 0.06 and adjusted using a Cosine Annealing scheduler. The optimizer is SGD, with a momentum of 0.9 and a weight decay of $5 \times 10^{-4}$. All experiments are conducted on four NVIDIA RTX A6000s and one NVIDIA GeForce RTX 4090. In all experiments, we set $M = 2$ and $N = 6$. Both $T^g$ and $T^l$ are composed of random cropping, color jitter, random flipping, and random grayscale, with respective cropping ranges of (0.4, 1.0) and (0.05, 0.4).

Furthermore, the settings for other parameters are as follows:

**Experiment on CIFAR10.** We set $k_{pub} = 256$, $k_{pvt} = 128$, $K = 30$ and $a = 10000$.

**Experiment on ImageNette.** We set $k_{pub} = k_{pvt} = 32$, $K = 50$ and $a = 0.1$.

**Experiment on ImageNet.** We set $k_{pub} = k_{pvt} = 32$ and $K = 50$ and $a = 1$.

### A.3   DETAILED EXPERIMENTAL RESULTS ON CIFAR10

This section presents the experimental results ($p$-values) of several baselines and our method on CIFAR10. $\mathcal{D}_{pub}$ is CIFAR10-1. The results are shown in Table 10, Table 11, and Table 12, respectively. '$\mathcal{D}_{sus}$' is the dataset used to pre-train $\mathcal{M}_{sus}$. 'CIFAR10-1' and 'CIFAR10-2' are the two non-overlapping halves of CIFAR10 training set after a random split. Each value is an average of 3 trials. Note that when $\mathcal{D}_{sus}$ is CIFAR10-1 or CIFAR10, the suspect used $\mathcal{D}_{pub}$, and this scenario is illegal, so $p$ should be less than 0.05. However, when $\mathcal{D}_{sus}$ is CIFAR10-2 or SVHN, the suspect did not use $\mathcal{D}_{pub}$, and this scenario is legal, so $p$ should be greater than 0.05. The illegal ($p$ should be less than 0.05) and legal ($p$ should be greater than 0.05) scenarios correspond to the pink and green areas in Table 10, Table 11, and Table 12, respectively.

## A.4 DETAILED EXPERIMENTAL RESULTS ON IMAGENETTE

This section presents the experimental results ($p$-values) of several baselines and our method on ImageNette. $\mathcal{D}_{pub}$ is ImageNette-1. The results are shown in Table 13, Table 14, and Table 15, respectively. 'ImageNette-1' and 'ImageNette-2' are the two non-overlapping random halves of ImageNette training set. Each value is an average of 3 trials. Note that when $\mathcal{D}_{sus}$ is ImageNette-1 or ImageNette, the suspect is illegal, so $p$ should be less than 0.05. However, when $\mathcal{D}_{sus}$ is ImageNette-2 or SVHN, the suspect is legal, so $p$ should be greater than 0.05. The illegal ($p$ should be less than 0.05) and legal ($p$ should be greater than 0.05) scenarios correspond to the pink and green areas in Table 13, Table 14, and Table 15, respectively.

## A.5 THE IMPACT OF SHADOW MODEL'S TRAINING HYPERPARAMETER

In the real world, the training hyperparameters of shadow models and suspicious models are often different. We analyzed whether these differences would affect our method. Specifically, we set different batch size (32 for the shadow model and 64 for the suspicious model), learning rate (0.01 for the shadow model and 0.06 for the suspicious model), and weight decay (1e-4 for the shadow model and 5e-4 for the suspicious model) for the shadow model compared to the suspicious model. As shown in Table 9, our method demonstrates good robustness to the training hyperparameter settings of the shadow model.

Table 9: The impact of shadow model's training hyperparameters on our method. Model is ResNet18. Both $\mathcal{D}_{pub}$ and $\mathcal{D}_{sus}$ are ImageNette. $\mathcal{D}_{sdw}$ is SVHN.

| Method | All Same | Different Batch Size | Different Learning Rate | Different Weight Decay |
|--------|----------|----------------------|-------------------------|------------------------|
| SimCLR | $10^{-11}$ | $10^{-6}$ | $10^{-10}$ | $10^{-5}$ |
| BYOL | $10^{-10}$ | $10^{-4}$ | $10^{-3}$ | $10^{-4}$ |
| SimSiam | $10^{-5}$ | $10^{-4}$ | $10^{-3}$ | $10^{-3}$ |

## A.6 THE DETAILS OF FINE-TUNING THE PRE-TRAINED MODEL

We fine-tuned the encoder on CIFAR10/CIFAR100 using a learning rate of 0.001, a batch size of 512, a weight decay of 5e-4, and the SGD optimizer with a momentum of 0.9. The pre-trained ResNet50 on ImageNet is sourced from MMSelfSup[3].

---

[3]https://mmselfsup.readthedocs.io/en/latest/model_zoo.html

Table 10: $p$-values for each scenario of DI4SSL on CIFAR10.

| Alg | Method | Model | $\mathcal{D}_{sus}$ | | | |
|---|---|---|---|---|---|---|
| | | | CIFAR10-1 | CIFAR10 | CIFAR10-2 | SVHN |
| DI4SSL | SimCLR | VGG16 | 0.01 | 0.43 | 0.54 | 0.42 |
| | | ResNet18 | 0.64 | 0.97 | 0.71 | 0.62 |
| | BYOL | VGG16 | $10^{-4}$ | $10^{-16}$ | 0.42 | 0.60 |
| | | ResNet18 | 0.99 | $10^{-5}$ | 0.44 | 0.73 |
| | SimSiam | VGG16 | 0.99 | 0.99 | 0.28 | 0.50 |
| | | ResNet18 | 0.79 | 0.93 | 0.41 | 0.89 |
| | MoCo v3 | VGG16 | 1 | 0.99 | 0.27 | 0.06 |
| | | ResNet18 | 1 | 1 | 0.46 | 0.82 |
| | DINO | ViT-T/4 | 0.90 | 0.52 | 0.75 | 0.55 |
| | | ViT-S/4 | 0.85 | 0.17 | 0.67 | 0.94 |

Table 11: $p$-values for each scenario of EncoderMI on CIFAR10.

| Alg | Method | Model | $\mathcal{D}_{sus}$ | | | |
|---|---|---|---|---|---|---|
| | | | CIFAR10-1 | CIFAR10 | CIFAR10-2 | SVHN |
| EncoderMI | SimCLR | VGG16 | 0 | 0 | 0 | 0 |
| | | ResNet18 | 0 | 0 | 0 | 0 |
| | BYOL | VGG16 | 0 | 0 | 0 | 0 |
| | | ResNet18 | 0 | 0 | 0 | $10^{-43}$ |
| | SimSiam | VGG16 | 0 | 0 | 0 | 0 |
| | | ResNet18 | 0 | 0 | $10^{-13}$ | 0 |
| | MoCo v3 | VGG16 | 0 | 0 | 0 | 0 |
| | | ResNet18 | 0 | 0 | 0 | $10^{-74}$ |
| | DINO | ViT-T/4 | 0.99 | 0.94 | 1 | 0.80 |
| | | ViT-S/4 | 0.99 | 0.99 | 1 | 1 |

Table 12: $p$-values for each scenario of our method on CIFAR10.

| Alg | Method | Model | $\mathcal{D}_{sus}$ | | | |
|---|---|---|---|---|---|---|
| | | | CIFAR10-1 | CIFAR10 | CIFAR10-2 | SVHN |
| Ours | SimCLR | VGG16 | $10^{-16}$ | $10^{-12}$ | 0.75 | 0.24 |
| | | ResNet18 | $10^{-12}$ | $10^{-8}$ | 0.29 | 0.32 |
| | BYOL | VGG16 | $10^{-20}$ | $10^{-18}$ | 0.91 | 0.64 |
| | | ResNet18 | $10^{-17}$ | $10^{-11}$ | 0.55 | 0.33 |
| | SimSiam | VGG16 | $10^{-4}$ | $10^{-6}$ | 0.99 | 0.99 |
| | | ResNet18 | $10^{-11}$ | $10^{-5}$ | 0.47 | 0.87 |
| | MoCo v3 | VGG16 | $10^{-11}$ | $10^{-14}$ | 0.91 | 0.76 |
| | | ResNet18 | $10^{-4}$ | $10^{-3}$ | 0.84 | 0.76 |
| | DINO | ViT-T/4 | 0.03 | 0.01 | 0.63 | 0.67 |
| | | ViT-S/4 | $10^{-7}$ | $10^{-7}$ | 0.43 | 0.62 |

Table 13: $p$-values for each scenario of DI4SSL on ImageNette.

| Alg | Method | Model | $\mathcal{D}_{sus}$ | | | |
|---|---|---|---|---|---|---|
| | | | ImageNette-1 | ImageNette | ImageNette-2 | ImageWoof |
| DI4SSL | SimCLR | VGG16 | 0.01 | 0.43 | 0.54 | 0.42 |
| | | ResNet18 | 0.64 | 0.97 | 0.71 | 0.62 |
| | BYOL | VGG16 | $10^{-4}$ | $10^{-16}$ | 0.42 | 0.60 |
| | | ResNet18 | 0.99 | $10^{-5}$ | 0.44 | 0.73 |
| | SimSiam | VGG16 | 0.99 | 0.99 | 0.28 | 0.50 |
| | | ResNet18 | 0.79 | 0.93 | 0.41 | 0.89 |
| | MoCo v3 | VGG16 | 1 | 0.99 | 0.27 | 0.06 |
| | | ResNet18 | 1 | 1 | 0.46 | 0.82 |
| | DINO | ViT-T/4 | 0.90 | 0.52 | 0.75 | 0.55 |
| | | ViT-S/4 | 0.85 | 0.17 | 0.67 | 0.94 |

Table 14: $p$-values for each scenario of EncoderMI on ImageNette.

| Alg | Method | Model | $\mathcal{D}_{sus}$ | | | |
|---|---|---|---|---|---|---|
| | | | ImageNette-1 | ImageNette | ImageNette-2 | ImageWoof |
| EncoderMI | SimCLR | VGG16 | 0 | 0 | 0 | 0 |
| | | ResNet18 | 0 | 0 | 0 | 0 |
| | BYOL | VGG16 | 0 | 0 | 0 | 0 |
| | | ResNet18 | 0 | 0 | 0 | $10^{-43}$ |
| | SimSiam | VGG16 | 0 | 0 | 0 | 0 |
| | | ResNet18 | 0 | 0 | $10^{-13}$ | 0 |
| | MoCo v3 | VGG16 | 0 | 0 | 0 | 0 |
| | | ResNet18 | 0 | 0 | 0 | $10^{-74}$ |
| | DINO | ViT-T/4 | 0.99 | 0.94 | 1 | 0.80 |
| | | ViT-S/4 | 0.99 | 0.99 | 1 | 1 |

Table 15: $p$-values for each scenario of our method on ImageNette.

| Alg | Method | Model | $\mathcal{D}_{sus}$ | | | |
|---|---|---|---|---|---|---|
| | | | ImageNette-1 | ImageNette | ImageNette-2 | ImageWoof |
| Ours | SimCLR | VGG16 | $10^{-16}$ | $10^{-12}$ | 0.75 | 0.24 |
| | | ResNet18 | $10^{-12}$ | $10^{-8}$ | 0.29 | 0.32 |
| | BYOL | VGG16 | $10^{-20}$ | $10^{-18}$ | 0.91 | 0.64 |
| | | ResNet18 | $10^{-17}$ | $10^{-11}$ | 0.55 | 0.33 |
| | SimSiam | VGG16 | $10^{-4}$ | $10^{-6}$ | 0.99 | 0.99 |
| | | ResNet18 | $10^{-11}$ | $10^{-5}$ | 0.47 | 0.87 |
| | MoCo v3 | VGG16 | $10^{-11}$ | $10^{-14}$ | 0.91 | 0.76 |
| | | ResNet18 | $10^{-4}$ | $10^{-3}$ | 0.84 | 0.76 |
| | DINO | ViT-T/4 | 0.03 | 0.01 | 0.63 | 0.67 |
| | | ViT-S/4 | $10^{-7}$ | $10^{-7}$ | 0.43 | 0.62 |

### A.7 THE IMPACT OF SAMPLING SIZE

We also evaluate the performance of our method using different amounts of data. Specifically, we conduct verification by selecting different sampling size $k_{pub}$ and $k_{pvt}$. We conduct experiments using pre-trained encoders on ImageNet, with the encoder architecture being ResNet50. Figure 5 shows that, across various contrastive learning methods, the effectiveness of our method improves as $k_{pub}$ and $k_{pvt}$ increase. This is because larger sampling size better represent the distribution of the dataset, making the contrastive relationship gap of the encoder more pronounced. Note that in this scenario, $\mathcal{D}_{sus}$ includes $\mathcal{D}_{pub}$ are both ImageNet, and $\mathcal{D}_{sus}$ includes $\mathcal{D}_{pub}$, so the $p$-values should be less than 0.05.

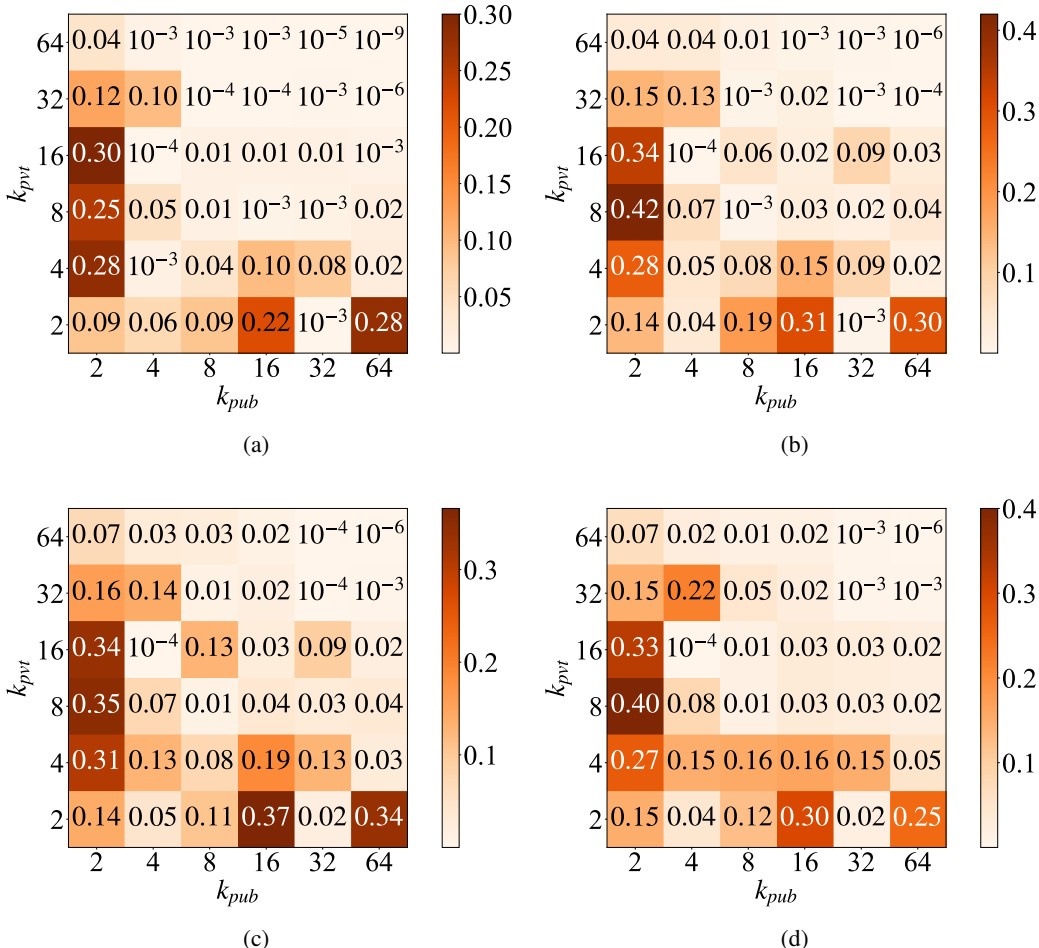

Figure 5: The $p$-values obtained using pre-trained ResNet50 on ImageNet with different $k_{pub}$ and $k_{pvt}$ values. Each heatmap corresponds to the results of different training algorithms. Figure 5a: SimCLR, Figure 5b: BYOL, Figure 5c: SimSiam, and Figure 5d: MoCo v3.

## A.8 THE IMPACT OF GLOBAL AND LOCAL AUGMENTATION NUMBER

We evaluate the effectiveness of our method under different numbers of global augmentations $M$ and local augmentations $N$. We conduct experiments using pre-trained ResNet50 on ImageNet. Figure 6 shows that, the performance of our method improves as $M$ and $N$ increase. This is because a greater number of augmentations provides more information to the encoder, thereby amplifying the contrastive relationship gap. Note that in this scenario, $\mathcal{D}_{sus}$ includes $\mathcal{D}_{pub}$ are both ImageNet, and $\mathcal{D}_{sus}$ includes $\mathcal{D}_{pub}$, so the $p$-values should be less than 0.05.

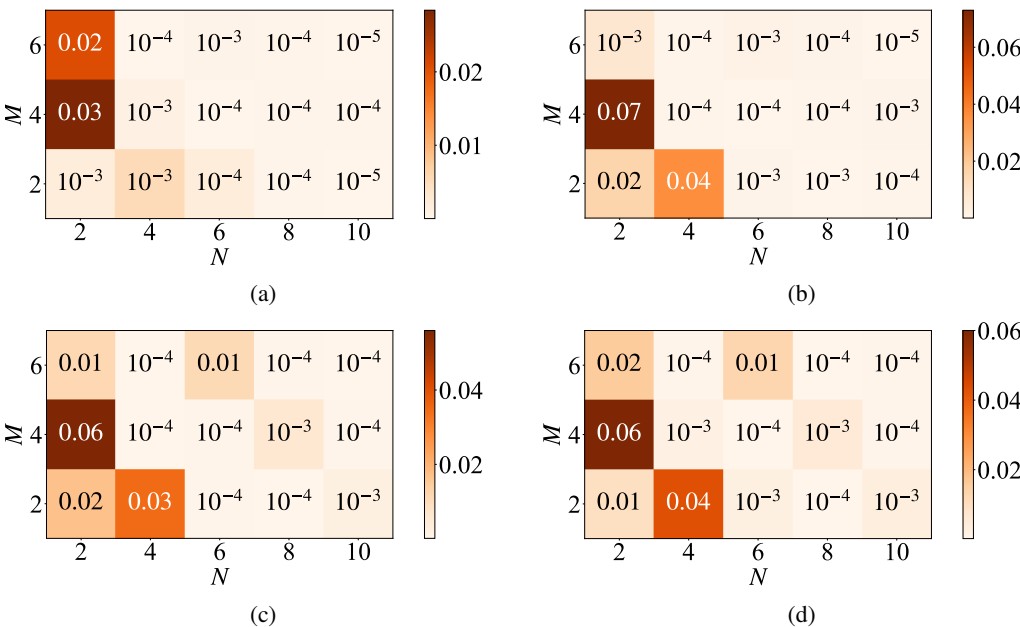

Figure 6: The $p$-values obtained using pre-trained ResNet50 on ImageNet with different $M$ and $N$ values. Each heatmap corresponds to the results of different training algorithms. Figure 6a: SimCLR, Figure 6b: BYOL, Figure 6c: SimSiam, and Figure 6d: MoCo v3.

## A.9 THE IMPACT OF SHADOW DATASET AND HYPERPARAMETER $a$

We investigate the impact of the shadow dataset $\mathcal{D}_{sdw}$ and the hyperparameter $a$ on our method. As shown in Figure 7, the setting of $a$ affects the validation results of our method. This effect is related to the distributions of $\mathcal{D}_{pub}$ and $\mathcal{D}_{sdw}$ and is not fixed. This indicates that the defender need to set appropriate $a$ based on his actual situation.

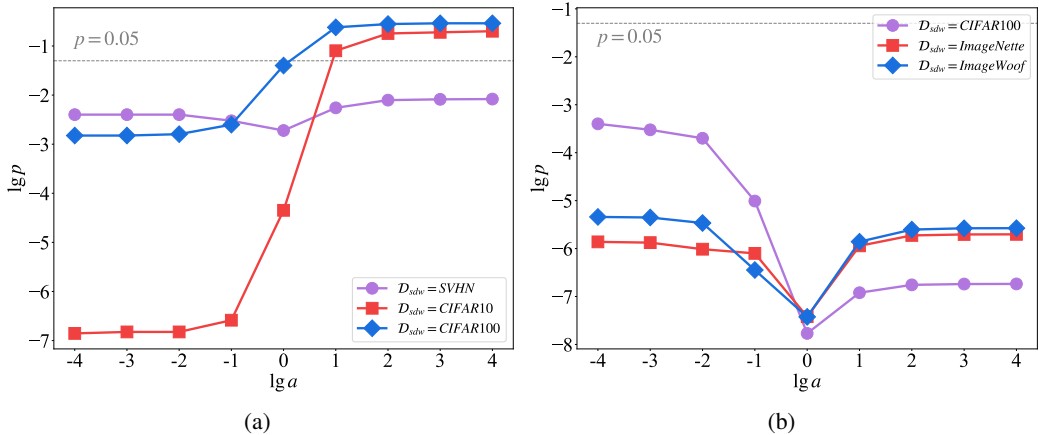

Figure 7: The impact of shadow dataset and hyperparameter $a$ on our method. The left figure represent the cases where $\mathcal{D}_{pub}$ is ImageNet and $\mathcal{M}_{sus}$ is a pre-trained ResNet50 using SimCLR, and in the right figure, $\mathcal{D}_{pub}$ and $\mathcal{M}_{sus}$ are CIFAR10 and a pre-trained ResNet18 using SimCLR, respectively.

## A.10 THE IMPACT OF EARLY STOPPING ON OUR METHOD

The early stopping technique can terminate model training prematurely, which may result in less pronounced contrastive relationship gap. To investigate the impact of early stopping on our method, we specifically set the patience of early stopping (the maximum number of epochs allowed to continue training when the K-Nearest Neighbors accuracy on the validation set does not improve significantly over multiple consecutive epochs) to 15 and 30, respectively. We then calculated the $p$-values of the trained models using the same method, as shown in Table 16. Both the datasets of defender and suspect are CIFAR10, meaning $p$-value should be less than 0.05. Self-supervised method is SimCLR. The shadow model is a ResNet18 pre-trained on ImageWoof using SimCLR. The results demonstrate that our method remains effective even under early stopping conditions.

Table 16: The results ($p$-values) of our method on suspicious models that used early stopping. The datasets of defender and suspect are both CIFAR10, making the suspect's behavior illegal, so the $p$-values should be less than 0.05.

| Model | w/o Early Stopping | w/ Early Stopping (patience=15) | w/ Early Stopping (patience=30) |
|---|---|---|---|
| ResNet18 | $10^{-12}$ | 0.01 | $10^{-4}$ |
| VGG16 | $10^{-11}$ | $10^{-4}$ | $10^{-5}$ |

## A.11   THE COMPARISON OF OUR METHOD WITH THE WATERMARK-BASED METHOD

Currently, there is no watermark-based dataset ownership verification method for pre-trained encoders, so we adapt CTRL Li et al. (2023a), the current state-of-the-art backdoor attack for self-supervised encoders, into a watermark-based dataset ownership verification method. Specifically, prior to the release of public dataset, we inject the the CTRL trigger as watermark into a small subset of the data. During the verification phase, we input both watermarked and non-watermarked images into the suspicious encoder. If the representations of the watermarked images are significantly more similar to each other than those of the non-watermarked images, we can conclude that the suspicious encoder was pre-trained using the public dataset. Table 17 indicates that although methods based on watermark can accurately identify cases where public datasets have been stolen, they also wrong the innocent suspect.

Table 17: The comparison of our method with watermark-based method ($\mathcal{D}_{pub}$ is CIFAR10). '$\mathcal{D}_{sus}$' is the dataset used to pre-train $\mathcal{M}_{sus}$, 'Alg' is the dataset ownership verification method. Each value is an average of 3 trials. The illegal ($p$ should be less than 0.05) and legal ($p$ should be greater than 0.05) scenarios correspond to the pink and green areas.

| Alg | Method | Model | $\mathcal{D}_{sus}$ | | | |
| --- | --- | --- | --- | --- | --- | --- |
| | | | CIFAR10 | SVHN | ImageNette | ImageWoof |
| DOV-CTRL | SimCLR | VGG16 | $10^{-317}$ | $10^{-4}$ | $10^{-6}$ | $10^{-49}$ |
| | | ResNet18 | $10^{-287}$ | $10^{-6}$ | 0.04 | 0.19 |
| | BYOL | VGG16 | 0 | 0.02 | 0.57 | 0.47 |
| | | ResNet18 | 0 | 0.29 | 0.52 | 0.04 |
| | SimSiam | VGG16 | $10^{-19}$ | 0.17 | $10^{-8}$ | $10^{-4}$ |
| | | ResNet18 | $10^{-290}$ | 0.25 | 0.18 | 0.31 |
| Ours | SimCLR | VGG16 | $10^{-10}$ | 0.45 | 0.67 | 0.99 |
| | | ResNet18 | $10^{-7}$ | 0.41 | 0.19 | 0.16 |
| | BYOL | VGG16 | $10^{-13}$ | 0.84 | 0.85 | 0.90 |
| | | ResNet18 | $10^{-9}$ | 0.61 | 0.59 | 0.59 |
| | SimSiam | VGG16 | $10^{-4}$ | 0.99 | 0.98 | 0.98 |
| | | ResNet18 | $10^{-4}$ | 0.88 | 0.36 | 0.37 |

## A.12   THE PERFORMANCE OF OUR METHOD ON MAE

We also conduct experiments using encoders pre-trained with methods other than contrastive learning. We select Masked Autoencoder (MAE) He et al. (2022) for experimentation, which is a representative method of Masked Image Modeling (MIM). Specifically, we use pre-trained models on ImageNet from the official MAE repository[4], with encoder architectures ViT-B/16 and ViT-L/16. Additionally, to better adapt the encoders pre-trained with MIM, we incorporate random masking into our multi-scale augmentation. According to the experimental results presented in Table 18, our method still didn't perform well despite targeted improvement. We will address the DOV issue of MIM pre-trained models in our future work.

Table 18: The results ($p$-values) of our method on MAE ($\mathcal{D}_{pub}$ is ImageNet). '$\mathcal{D}_{sus}$' is the dataset used to pre-train $\mathcal{M}_{sus}$. Each value is an average of 3 trials. Note that in this scenario, $\mathcal{D}_{sus}$ and $\mathcal{D}_{pub}$ are both ImageNet and $\mathcal{D}_{sus}$ includes $\mathcal{D}_{pub}$, making the suspect's behavior illegal, so the $p$-values should be less than 0.05.

| Method | Model | Ours + Random Masking |
| --- | --- | --- |
| MAE | ViT-B/16 | 0.75 |
| | ViT-L/16 | 0.44 |

---

[4]https://github.com/facebookresearch/mae

## A.13 VISUALIZATION RESULTS

We present the visualization results of our method on ImageNette. Specifically, $\mathcal{D}_{pub}$ is set as ImageNette, and the shadow model is a ResNet18 trained on SVHN using SimCLR. We calculated the contrastive relationship gap $d$ of the shadow model and suspicious models trained on different datasets and visualized the comparison. When the suspicious model is pre-trained on $\mathcal{D}_{pub}$, it is considered illegal, and the contrastive relationship gap $d$ should be significantly higher than that of the shadow model. Conversely, if the suspicious model is legitimate, the two contrastive relationship gaps should be similar. As shown in Figure 8, when the suspicious model is illegal, its contrastive relationship gap is significantly higher than that of the shadow model. When the suspicious model is legitimate, the two contrastive relationship gaps are close. This observation aligns with our previous findings.

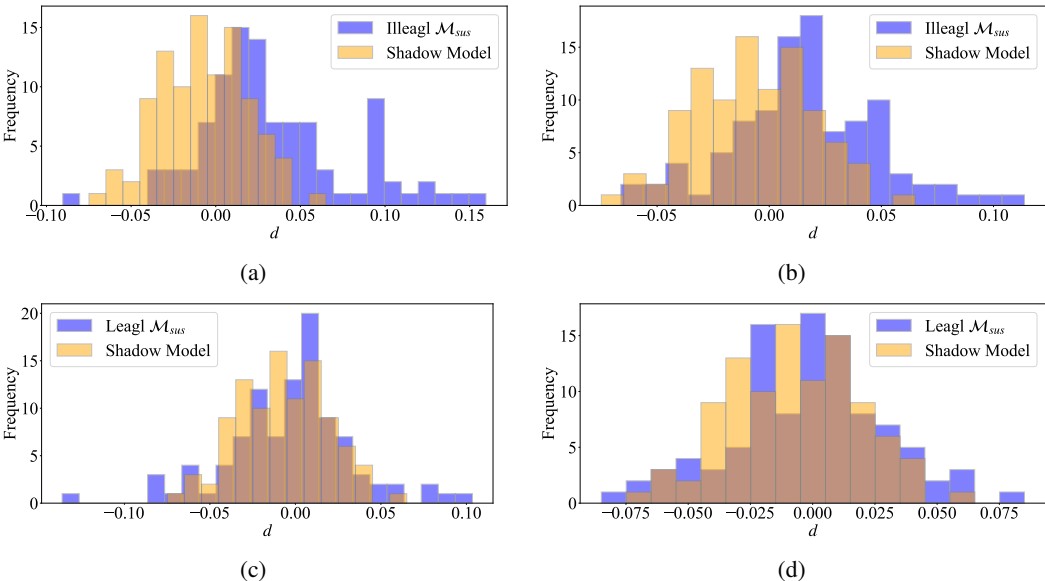

Figure 8: The contrastive relationship gap $d$ of the shadow model and suspicious models trained on different datasets. Each subplot corresponds to a different suspicious model. Figure 8a: suspicious model is a ResNet18 trained on ImageNette using SimCLR, Figure 8b: suspicious model is a ResNet18 trained on ImageNette using SimSiam, Figure 8c: suspicious model is a ResNet18 trained on ImageWoof using SimCLR, and Figure 8d: suspicious model is a ResNet18 trained on ImageWoof using SimSiam.

