# OpenReview forum: "Dataset Ownership Verification in Contrastive Pre-trained Models"
_ICLR.cc/2025/Conference — ICLR 2025 Poster_

### Official Review · Reviewer_DVkE · 2024-10-24

**Soundness:** 3
**Presentation:** 3
**Contribution:** 3
**Rating:** 8
**Confidence:** 3

**Summary:**

In this paper, the authors investigate the challenge of protecting high-quality open-source datasets from misuse by individuals who lack ownership verification. To address this issue, they propose a method that exploits a natural property of contrastive learning: the distinct distances between seen and unseen examples. Specifically, they demonstrate that these distances often appear significantly larger for unseen examples compared to their seen counterparts. The authors conduct an extensive set of experiments with various models and datasets to evaluate the efficacy of their proposed approach, which yields substantial improvements over existing baselines.

**Strengths:**

1. The research topic is important.
2. The authors conduct many experiments.
3. The performance is strong compared with baselines.

**Weaknesses:**

1. While the authors present distance metrics for dsus and dsdw, I believe it would be beneficial to include some visualizations.
2. Contrastive learning is currently a hot research area in computer vision, but the proposed methods appear to be limited to it, which may restrict their broader applicability.
3. The distances between examples are influenced by many factors beyond seen and unseen examples, including generalization capabilities and augmentations. I have concerns that the results presented may not fully support the claims.

**Questions:**

1. Does the proposed method work with the CLIP model, which also utilizes contrastive learning for pre-training?
2. Are the findings influenced by the number of training classes and examples? For instance, if Msus has 10k classes and Msdw has only 10, do the proposed methods work as expected?
3. How sensitive is our method to the choice of augmentations? Specifically, we consider scenarios where new augmentation techniques are introduced during evaluation but were not present during training of Msus. In such cases, the distances between outputs from Msus may increase, even when it used public data?

---

> ### Author Response · Authors · 2024-11-21
> **Rebuttal (1/3)**
>
> We are grateful for your detailed feedback and careful review. It is a valuable opportunity for us to address your concerns.
>
> > **W1: It would be beneficial to include some visualizations.**
>
> Thank you for your suggestion. We agree with your perspective that some visualization results can help readers better understand the paper. To address this, we have added visualizations of some representative examples in Appendix A.13. We present the visualization results of our method on ImageNette. Specifically, $ \mathcal{D}\_{pub} $ is set as ImageNette, and the shadow model is a ResNet18 trained on SVHN using SimCLR. We calculated the contrastive relationship gap $ d $ of the shadow model and suspicious models trained on different datasets and visualized the comparison.
> When the suspicious model is pre-trained on $ \mathcal{D}\_{pub} $, it is considered illegal, and the contrastive relationship gap $ d $ should be significantly higher than that of the shadow model. Conversely, if the suspicious model is legitimate, the two contrastive relationship gaps should be similar.
>
> For details, please refer to Appendix A.13 in the attached PDF.
>
> > **W2: The proposed method's applicability is limited to contrastive learning models.**
>
> Thank you for your feedback. We agree that broadening the scope of the method to encompass more self-supervised techniques could amplify its utility and influence. This paper, serving as a pioneering endeavor towards this direction, focuses on dataset ownership verification within contrastive pre-trained models, one of the most representative and well-developed self-supervised approaches. We plan to tackle the issue of DOV for more self-supervised pre-trained models in our future research endeavors.

---

> ### Author Response · Authors · 2024-11-21
> **Rebuttal (2/3)**
>
> > **W3&Q3: The distances between examples are influenced by many factors beyond seen and unseen examples, including generalization capabilities and augmentations.**
>
> Thanks for the nice question! The generalization ability of a model is primarily influenced by two factors [1]: the model architecture and the distributional differences between the training and testing datasets. The experiments in our paper demonstrate the robustness of our method to both aspects, as detailed below:
>
> - Regarding model architecture, we validate the robustness of our method by employing different architectures in our experiments, including convolutional neural networks and transformers.
> - Regarding the distributional differences between the training and testing datasets, we designed our experiments by splitting CIFAR10 into two disjoint subsets, CIFAR10-1 and CIFAR10-2, which share almost identical data distributions. However, as shown in the experimental results in Figure 3, even in this extreme scenario (where the distributions of the datasets used by the suspect and the defender are nearly identical, but non-overlapping), our method still avoids falsely accusing innocent suspicious models.
>
> For data augmentation, we considered the following two scenarios:
>
> - The defender and the suspect use different augmentation techniques.
>
>     For different augmentation strategies, we conducted additional experiments on CIFAR10. Both the datasets of defender and suspect are CIFAR10. Specifically, when training the suspicious model, we removed one of the augmentation strategies from cropping, flipping, jitter, and grayscale. The K-Nearest Neighbors (KNN) accuracy on the testing set and $p$-values are as follows. The model and contrastive learning method used are ResNet18 and SimCLR, respectively.
>
>     When the suspicious model is trained without using cropping, it evaded our detection ($p>0.05$). This is because random cropping is one of the augmentation strategies that has the greatest impact on model performance [2]. Removing cropping significantly degrades the model's performance, resulting in less distinguishable contrastive relationships, thereby causing the detection to fail.
>
> |  | All Same | w/o Cropping | w/o Flipping | w/o Jitter | w/o Grayscale |
> | :---- | :---- | :----  | :---- | :----  | :---- |
> | $p$  | $10^{-12}$ | 0.88 | $10^{-4}$ | $10^{-6}$ | $10^{-5}$ |
>
> - The defender and the suspect use different augmentation hyperparameters.
>
>     For differently parametrized augmentations, here we conducted supplementary experiments on ImageNette with different cropping, flipping, jitter and grayscale. Both the datasets of defender and suspect are ImageNette. Specifically, the changes to the shadow model’s augmentation parameters are as follows:
>
>     1) The global/local cropping size is changed from (0.4, 1.0)/(0.05,0.4) to (0.6, 1.0)/(0.05,0.6);
>     2) The probability of random flipping is changed from 0.5 to 0.2;
>     3) The jitter parameters are changed from (0.4, 0.4, 0.4, 0.1) to (0.2, 0.2, 0.2, 0.2);
>     4) The probability of grayscale is changed from 0.2 to 0.5.
>
>     The $p$-value results listed below show that although different parameters may increase the $ p $-value, our method remains effective. Model is ResNet18:
>
> | Self-supervised Method | All Same | Different Cropping | Different Flipping | Different Jitter | Different Grayscale |
> | :---- | :---- | :----  | :---- | :----  | :---- |
> | SimCLR  | $10^{-11}$ | $10^{-5}$ | $10^{-11}$ | $10^{-10}$ | $10^{-11}$ |
> | BYOL  | $10^{-10}$ | $10^{-5}$ | $10^{-9}$ | $10^{-9}$ | $10^{-10}$ |
> | SimSiam  | $10^{-5}$ | $10^{-3}$ | $10^{-5}$ | $10^{-5}$ | $10^{-5}$ |
>
> ### Reference
> [1] Ben-David S, Blitzer J, Crammer K, et al. Analysis of representations for domain adaptation[J]. Advances in neural information processing systems, 2006, 19.
>
> [2] Chen T, Kornblith S, Norouzi M, et al. A simple framework for contrastive learning of visual representations[C]//International conference on machine learning. PMLR, 2020: 1597-1607.

---

> ### Author Response · Authors · 2024-11-21
> **Rebuttal (3/3)**
>
> > **Q1: Does the proposed method work with the CLIP model?**
>
> Thank you for posing this insightful question. CLIP is jointly pre-trained on 400 million pairs of (image, text) collected from the internet. OpenAI has publicly provided the image encoder and text encoder. We treat the CLIP image encoder with a ViT-B/32 architecture as the suspicious encoder. Specifically, given a dataset, our goal is to use our method to infer whether it has been utilized by CLIP.
>
> Considering the scenario where the training dataset of CLIP is unknown, we follow the experimental setup in [3]. Specifically, we use the class names of CIFAR100 as keywords and collect images using Bing Image Search. For each keyword, we collect 10 images, resulting in a total of 1,000 images. These images are regarded as the potential dataset used to train CLIP, corresponding to $\mathcal{D}_{pub}$ in our setting (note that we cannot guarantee every image was used to train CLIP, but as common categories, these images are likely to have been collected and used).
>
> To construct a dataset not used by CLIP, we further collect 2,000 images using the keywords from Bing Search. These images are randomly paired into 1,000 pairs, each consisting of two images from different categories. For each pair, we resize the two images to the same size and then concatenate them to form a new image, resulting in a total of 1,000 images. This dataset is regarded as non-potential dataset (not used by CLIP), corresponding to $\mathcal{D}_{pvt}$ in our setting.
> Additionally, we use a ResNet18 trained via SimCLR as the shadow encoder and test our method on different shadow datasets. The results are summarized below, where each $p$-value is the average of three experiments:
>
> | $\mathcal{D}_{sdw}$ | SVHN | CIFAR10 |
> | :---- | :---- | :----  |
> | $p$ | $10^{-4}$ | $10^{-18}$ |
>
> The results in the table indicate that under the setting in [3] (where CLIP may have been trained using the potential dataset), the potential dataset is more likely to have been used to train CLIP compared to the non-potential dataset, which aligns with our intuition.
>
> ### Reference
> [3] Liu H, Jia J, Qu W, et al. Encodermi: Membership inference against pre-trained encoders in contrastive learning[C]//Proceedings of the 2021 ACM SIGSAC Conference on Computer and Communications Security. 2021: 2081-2095.
>
> > **Q2: If $\mathcal{M}\_{sus}$ has 10k classes and $\mathcal{M}\_{sdw}$ has only 10, do the proposed methods work as expected?**
>
> Thank you for your feedback. Our method is unaffected by the number of classes in the shadow dataset or the suspect’s dataset. In the CIFAR10 experiment, the shadow dataset (CIFAR100) contains 100 classes, while the suspect’s dataset (CIFAR10) has only 10 classes. Conversely, in the ImageNet experiment, the shadow dataset (SVHN) has only 10 classes, whereas the suspect’s dataset (ImageNet) contains 1000 classes. In both experiments, the number of classes in the datasets varies significantly, yet our method remains effective. This is because our approach relies on the unary relationships of individual samples and the binary relationships between different samples, regardless of the classes of these samples. In other words, the mechanism of our method is not directly related to the number of classes in the dataset.

---

> > ### Comment · Reviewer_DVkE · 2024-11-25
> >
> > Thank you for taking the time to provide thoughtful responses to my questions. CIFAR10 is a simple dataset, whose differences in classes are dynamic. Due to the time, I can accept those results, and I've accordingly adjusted my evaluation to a positive mark.
> >
> > Furthermore, I noticed that the table headers in the rebuttal (Section 3/3) appear to be incorrect. Additionally, the value of 10^{-4} for the non-potential dataset is lower than the threshold of 0.05.

---

> ### Author Response · Authors · 2024-11-25
>
> Thank you for your feedback and increasing the score! Your suggestions have played a crucial role in enhancing our work.
>
> Additionally, regarding your concerns about the table headers in the CLIP experiment results, we will provide a more detailed explanation. In the CLIP experiments, we used the following datasets:
>
> - **Potential Dataset**: This dataset likely contains images that were used to train CLIP, equivalent to $\mathcal{D}_{pub}$ in our setup. It consists of 1,000 images collected from the internet by searching CIFAR100 categories.
> - **Non-Potential Dataset**: This dataset likely does not contain images used to train CLIP, equivalent to the test set of $\mathcal{D}_{pub}$ in our setup. It consists of 1,000 synthetic images, each created by merging two randomly collected images from the internet (also retrieved via CIFAR100 category searches).
> - **Shadow Dataset ($\mathcal{D}_{sdw}$)**: This dataset is used by the defender to train the shadow model. It can be any dataset different from $\mathcal{D}_{pub}$.
>
> In the table presenting the CLIP experiment results, the header refers to the shadow dataset $\mathcal{D}_{sdw}$. The results indicate that regardless of whether the defender uses CIFAR10 or SVHN as the shadow dataset for verification, our method consistently identifies that CLIP is more likely trained on the potential dataset compared to the non-potential dataset, which aligns with our intuition.
>
> In addition, the $p$-value is calculated for the potential dataset. If the $p$-value is less than 0.05, it indicates that the potential dataset was likely used for pretraining CLIP. Conversely, if the $p$-value is greater than 0.05, it suggests that the potential dataset was not used for training CLIP.
>
> Thank you again for your feedback! We hope our responses fully address your concerns.

---

> > ### Comment · Reviewer_DVkE · 2024-11-29
> >
> > Since non-potential datasets are randomly synthesized, their distances between classes should be significantly larger than those of potential datasets. This distinction holds even when potential datasets are not used.
> >
> > I would appreciate it if the authors could consider testing whether CLIP employs training data used in their evaluation of zero-shot tests, such as CIFAR10 and Food101. Furthermore, this technique would be significant for LLMs, whether those models employ test data or not.

---

> ### Author Response · Authors · 2024-11-29
>
> Thank you for your insightful feedback. Below, we provide further clarification on the two issues you raised.
>
> > **Non-potential datasets are randomly synthesized.**
>
> In response to your suggestions, we utilized real, non-synthetic images as samples for the non-potential datasets. Specifically, we conducted experiments using the following two datasets: **images captured with our mobile phone** and the **SODA-D dataset** [1].
>
> The first dataset comprises 500 photos taken with our phone, including images of people, landscapes, and other subjects. These images have not been publicly released and, as such, have not been pre-trained by CLIP. The second dataset, SODA-D, is a large-scale benchmark for Small Object Detection, featuring 24,828 meticulously curated and high-quality images from driving scenarios. Certain samples were collected from real-world scenes, and the dataset was released a year after CLIP, ensuring it was not part of CLIP’s pre-training corpus. Due to their privacy and release timing, both datasets are ideal as non-potential datasets.
>
> The results of our method on these non-potential datasets are as follows. The potential dataset remains the 1,000 images we sourced online based on CIFAR-100 categories. The shadow model is a ResNet18 pre-trained using SimCLR on SVHN.
>
> | Non-Potential Dataset          | Images Captured with Phone | SODA-D  |
> |--------------------------------|----------------------------|---------|
> | $p$                            | $10^{-25}$                 | $10^{-3}$|
>
> The results in the table again indicate that the potential dataset is more likely to have been used to train CLIP compared to the non-potential dataset.
>
> > **Does CLIP Use Training Data Employed in Zero-Shot Tests?**
>
> According to the CLIP paper [2], several datasets were selected for zero-shot evaluation, including CIFAR10, CIFAR100, Food101, and ImageNet. These datasets, collectively denoted as $\mathcal{D}\_{zs}$, were used solely for zero-shot testing in the original paper. As such, CLIP should not be pre-trained on the training sets of $\mathcal{D}\_{zs}$, implying that the $p$-value should exceed 0.05.
>
> In subsequent experiments, we employed a ResNet18 pre-trained using SimCLR on SVHN as the shadow model. Here, $\mathcal{D}\_{pub}$ represents the training set of $\mathcal{D}\_{zs}$ (equivalent to the earlier "potential dataset"), and $\mathcal{D}\_{pvt}$ represents the images captured with our phone or SODA-D (equivalent to the earlier "non-potential dataset"). The experimental results for each dataset are presented below.
>
> When $\mathcal{D}\_{pvt}$ comprises images captured with our phone:
>
> | $\mathcal{D}\_{zs}$ | CIFAR10 | CIFAR100 | Food101 | ImageNet |
> |---------------------|---------|----------|---------|----------|
> | $p$                | 0.99    | 0.99     | 1.00    | 0.68     |
>
> When $\mathcal{D}\_{pvt}$ is SODA-D:
>
> | $\mathcal{D}\_{zs}$ | CIFAR10 | CIFAR100 | Food101 | ImageNet |
> |---------------------|---------|----------|---------|----------|
> | $p$                | 1.00    | 0.99     | 1.00    | 1.00     |
>
> The results demonstrate that the $p$-values for all datasets exceed 0.05, affirming that CLIP does not use training data from the datasets in their zero-shot tests.
>
> We sincerely appreciate your interest in our work and hope that this response has adequately addressed your concerns.
>
> ### References
> [1] Cheng G, Yuan X, Yao X, et al. *Towards large-scale small object detection: Survey and benchmarks*. IEEE Transactions on Pattern Analysis and Machine Intelligence, 2023.
>
> [2] Radford A, Kim J W, Hallacy C, et al. *Learning transferable visual models from natural language supervision*. *Proceedings of the International Conference on Machine Learning*. PMLR, 2021: 8748-8763.

---

> > ### Comment · Reviewer_DVkE · 2024-12-02
> >
> > Based on the recent results, I revised my assessment and believe that this work will be highly valuable for the field.

---

> ### Author Response · Authors · 2024-12-02
>
> We are deeply grateful for your thoughtful review of our work and the generous rating you have bestowed upon it. It brings us great joy to know that we have successfully resolved your concerns.

---

### Official Review · Reviewer_KUSb · 2024-10-29

**Soundness:** 3
**Presentation:** 3
**Contribution:** 3
**Rating:** 6
**Confidence:** 4

**Summary:**

This work proposes the first dataset ownership verification method specifically for self-supervised pre-trained models using contrastive learning. The paper identifies significant variations in unary and binary instance relationships within embedding spaces when models are trained with specific datasets, compared to those trained without them. It introduces the concept of "contrastive relationship gap," a novel technique for verifying dataset ownership in contrastive pre-trained models. Extensive experiments demonstrate the approach's effectiveness, with a p-value significantly below 0.05, surpassing previous methods.

**Strengths:**

1. The paper introduces a novel method for dataset ownership verification (DOV) specifically tailored for contrastive pre-trained models, addressing a critical need in data rights protection.
2. The paper introduces the concept of "contrastive relationship gap," providing a clear technical approach to differentiate the model's performance on training and non-training datasets.
3. The method has been validated across multiple contrastive pre-trained models, including SimCLR, BYOL, SimSiam, MoCo v3, and DINO, demonstrating its broad applicability.
4. Experimental results show that the method can significantly outperform previous methodologies with a high probability of rejecting the null hypothesis (p-value well below 0.05).

**Weaknesses:**

1. As paper illustrated in limitations and conclusion, the method is primarily effective for encoders pre-trained with contrastive learning and may not perform well with other self-supervised learning pre-training methods like Masked Image Modeling (MIM).
2. The method lacks comparisons with enough baselines in the experimental section to clearly show its superiority.
3. CONTRASTIVE RELATIONSHIP GAP Part is difficult and too mathematical to understand. The authors could improve the writing to make it easier for the reader to understand.

**Questions:**

1. Could the authors provide both theoretical and experimental insights into the role of comparative learning within your approach?
2. Authors only compare the method with two baselines: DI4SSL and EncoderMI. Could the authors compare more recent related works?
3. Table.5 is missing EncoderMI time cost. Could author afford more information about time cost of other baselines?

---

> ### Author Response · Authors · 2024-11-21
> **Rebuttal (1/2)**
>
> We sincerely thank you for your comprehensive review and insightful suggestions. We are glad to have the opportunity to address your concerns.
>
> > **W1: The proposed method's applicability is limited to contrastive learning models.**
>
> Thank you for your feedback. We agree that broadening the scope of the method to encompass more self-supervised techniques could amplify its utility and influence. This paper, serving as a pioneering endeavor towards this direction, focuses on dataset ownership verification within contrastive pre-trained models, one of the most representative and well-developed self-supervised approaches. We plan to tackle the issue of DOV for more self-supervised pre-trained models in our future research endeavors.
>
> > **W2&Q2: Could the authors compare more recent related works?**
>
> Thank you for your question. In the appendix of original paper, we provide a comparison between our method and the CTRL [1] backdoor watermarking, one of the most advanced backdoor techniques designed for self-supervised models. Specifically, we injected CTRL triggers as watermarks into a small subset of data. During the verification phase, we input both watermarked and non-watermarked images into the suspicious model. If the representations of the watermarked images are significantly more similar than those of the non-watermarked images, we can conclude that the suspicious model was pre-trained on the protected dataset. The results in Appendix A.11 indicates that although methods based on CTRL can accurately identify cases where public datasets have been stolen, they also wrong the innocent suspect. For detailed results, please refer to Appendix A.11 in the attached PDF.
>
> Additionally, we utilized the state-of-the-art backdoor attack method for contrastive learning [2], named CorruptEncoder, to design the watermark. Specifically, we randomly selected 10 classes from ImageNet as the public dataset $\mathcal{D}_{pub}$, referred to as ImageNet-10, which is non-overlapping with ImageNette and ImageWoof. Following the setup in the original paper, we injected CorruptEncoder's backdoor watermark into 0.5% of the images in this dataset (poisoning rate of 0.5%). The verification process is identical to that of the CTRL-based method. The results of suspicious models pre-trained on different datasets are shown below. All suspicious models are ResNet18 pre-trained using SimCLR. The shadow model is a ResNet18 pre-trained on ImageWoof using SimCLR.
>
> | Method         | ImageNet-10 | CIFAR10 | CIFAR100 | ImageNette |
> | :----          | :----       | :----   | :----    | :----      |
> | CorruptEncoder | $10^{-35}$  | 0.02    | 0.93     | $10^{-24}$ |
> | Ours           | $10^{-14}$  | 0.99    | 0.74     | 0.71       |
>
> Note that the public dataset $\mathcal{D}_{pub}$ is ImageNet-10. Therefore, the $p$-values for suspicious models trained on ImageNet-10 should be less than 0.05, while those for models trained on other datasets should be greater than 0.05. The results indicate that although the CorruptEncoder watermarking method can identify illegal behavior, it also falsely accuses innocent suspicious models. This is similar to the results obtained from experiments using CTRL.
> This occurs because backdoor watermarks (triggers) are usually fixed patterns (e.g., specific small squares for CorruptEncoder or fixed-frequency noise for CTRL), to ensure the backdoor is successfully embedded into the model. These watermarked images tend to have similar features, leading the encoder to generate more similar representations for them compared to clean images, even if the encoder is trained on a non-watermarked dataset.
>
> ### Reference
> [1] Li C, Pang R, Xi Z, et al. An embarrassingly simple backdoor attack on self-supervised learning[C]//Proceedings of the IEEE/CVF International Conference on Computer Vision. 2023: 4367-4378.
>
> [2] Zhang J, Liu H, Jia J, et al. Data Poisoning based Backdoor Attacks to Contrastive Learning[C]//Proceedings of the IEEE/CVF Conference on Computer Vision and Pattern Recognition. 2024: 24357-24366.

---

> ### Author Response · Authors · 2024-11-21
> **Rebuttal (2/2)**
>
> > **W3: CONTRASTIVE RELATIONSHIP GAP Part is difficult and too mathematical to understand.**
>
> We apologize for any inconvenience caused. Below, I will provide a detailed explanation of the CONTRASTIVE RELATIONSHIP GAP Part.
>
> **Unary Relationship Gap:** The unary relationship gap $d_U$ is defined as $d_U=S_U^{pub}-S_U^{pvt}$. $S_U^{pub}$ is the unary relationship similarity of $\mathcal{D}\_{pub}$, and $S_U^{pvt}$ is the unary relationship similarity of $\mathcal{D}\_{pvt}$ (which is set as the testing set of $\mathcal{D}\_{pub}$ in the experiment section).
> The unary relationship similarity $S_U$ is composed of three parts: $S_U^{gg}$, $S_U^{gl}$, and $S_U^{ll}$. Here,
>
> - $S_U^{gg}$ represents the average cosine similarity of the representations of each sample, which undergoes two separate global augmentations.
>
> - $S_U^{gl}$ represents the average cosine similarity of the representations of each sample, which undergoes one global augmentation and one local augmentation.
>
> - $S_U^{ll}$ represents the average cosine similarity of the representations of each sample, which undergoes two separate local augmentations.
>
> **Binary Relationship Gap:** The binary relationship gap $d_B$ is defined as $d_B=S_B^{pub}-S_B^{pvt}$. $S_B^{pub}$ is the binary relationship similarity of $\mathcal{D}\_{pub}$, and $S_B^{pvt}$ is the binary relationship similarity of $\mathcal{D}\_{pvt}$.
> The binary relationship similarity $S_B^{pub}$ is composed of three parts: $S_B^{gg}$, $S_B^{gl}$, and $S_B^{ll}$. They represent the negative mean absolute error of two sets of cosine similarities. However, the contents of the two sets are different respectively.
>
> - $S_B^{gg}$: Two sets each contain the pairwise cosine similarities of the representations of global augmented samples.
> - $S_B^{gl}$: One set contains the pairwise cosine similarities of the representations of globally augmented samples, while the other set contains the pairwise cosine similarities of the representations of locally augmented samples.
> - $S_B^{ll}$: Two sets each contain the pairwise cosine similarities of the representations of local augmented samples.
>
> > **Q1: Could the authors provide both theoretical and experimental insights?**
>
> We apologize for any potential confusion caused. The two observations are extrapolated from the empirical findings, as depicted in Figure 1 of the main paper. The details are as follows:
>
> - Obversation 1 (Unary Relationship): the distinct augmented variants of the same sample within the training dataset are clustered more tightly, while the diverse augmented variants of the same sample within the test dataset are more scattered.
>
> - Obversation 2 (Binary Relationship): the pairwise cosine similarity between different augmentations of two samples in the training dataset shows smaller variation, whereas in the testing dataset, the pairwise cosine similarity between different augmentations of two samples exhibits larger variation.
>
> We calculate contrastive relationship gap using unary and binary relationships, which serve as the basis for verifying the dataset ownership.
>
> > **Q3: Could author afford more information about time cost of other baselines?**
>
> Thank you for your comment. As in our previous setup, we use a ResNet50 pre-trained on ImageNet as the suspicious model. Both the public dataset and the suspicious dataset are ImageNet, meaning $p$-value should be less than 0.05. The average time per verification and the corresponding performance of each method are as follows:
>
> | Method | Time Consumption | $p$ |
> | :---- | :---- | :----  |
> | EncoderMI  | 64s | 1 |
> | Ours  | 293s | $10^{-3}$ |
>
> Although EncoderMI is more time-efficient than our method, its performance is inferior. This is because EncoderMI only compares the similarity of different augmentations of the same sample (similar to the unary relationships in our method), which reduces computational cost but leads to the loss of binary relationship information between different samples.

---

> > ### Comment · Reviewer_KUSb · 2024-11-23
> >
> > Thanks to the author's reply, my confusion is solved so I am keeping the positive score.

---

> ### Author Response · Authors · 2024-11-23
>
> Thank you for your timely feedback. Your insights have been invaluable in helping us enhance the quality of our paper.

---

### Official Review · Reviewer_F2ST · 2024-10-29

**Soundness:** 2
**Presentation:** 1
**Contribution:** 2
**Rating:** 5
**Confidence:** 4

**Summary:**

This paper proposes a method for protecting datasets from infringement in contrastive learning scenarios by introducing two relationships: unary and binary. The unary relationship assesses the clustering ability of representations for augmentations of a known sample, while the binary relationship evaluates the separation between representations of two known samples. Using these properties, the authors define distance metrics to evaluate protected training data (known to both the suspect and defender) and secret data (unseen by the suspect but known to the defender). Since suspect models are overfitted to training data and have not seen the secret data, they exhibit different behavior for these two data types, which the method uses to identify suspect models.

**Strengths:**

This paper addresses an important and novel problem—dataset copyright protection in contrastive learning. The authors provide a comprehensive range of experiments, and the proposed method consistently demonstrates outstanding results across all tested settings.

**Weaknesses:**

I have several concerns:


1. The proposed unary and binary relationships align with the goals of contrastive learning, which promotes close representations for variants of the same sample and separation for different samples. The authors rely on overfitting to training data for verification, but as contrastive learning improves, this approach may be less effective. Enhanced contrastive learning might eventually generalize representations, clustering representations from single-sample into a single point, and representations obtained from $N$ distinct samples into $N$ separate points, even for unseen data. Though this is an idealized scenario, this aligns with the direction of contrastive learning research, so verification should be robust to and able to coexist with advances in contrastive learning.


2. The proposed method is similar to verification approaches in supervised learning that use differences in confidence scores or accuracies between training and test data. As Guo et al. [1] demonstrated, task performance differences between seen and unseen data are well-documented, but they are not commonly used for verification due to two primary reasons:
1) Future research is expected to reduce overfitting and improve generality, as noted in Comment 1.
2) An adversary could argue that similar samples exist in their training data by chance, complicating proof that observed low p-values are due to protected data.

To address these, verification metrics should be designed independently of task performance, ensuring that evidence cannot naturally occur by chance. Given this, the proposed method may lack admissible evidence of dataset infringement.

[1] Chuan Guo et al. On Calibration of Modern Neural Networks

3. Given the challenges noted above, many dataset protection methods in supervised learning use backdoor attacks or data poisoning. There are also backdoor attack studies specific to contrastive learning, such as Zhang et al. [2] and Carlini et al. [3]. The authors, however, only compare their method to a model protection technique and a unary-only method (e.g., EncoderMI). I suggest adding comparisons with established backdoor and data poisoning methods for contrastive learning.

[2] Zhang et al. Data Poisoning-based Backdoor Attacks to Contrastive Learning

[3] Carlini et al. Poisoning and Backdooring Contrastive Learning


4. Section 4.5.2 is critical, as contrastive learning is often used as a pretraining method, and adversaries are more likely to release fine-tuned models. Thus, verification post-fine-tuning is essential. However, this section only states that experiments were conducted, without presenting results in the main text. It references "Table 7 in Appendix A.6," which are outside the main manuscript. Ideally, essential content should be included in the main text, with the appendix for supplementary details. Additionally, details on the experiments are missing from the appendix, and Table 7 should include downstream performance results, as small learning rates could affect the reported outcomes. Additionally, Figure 3 occupies too much space; it would be better to reduce its size and include more analysis results directly in the main manuscript.


5. The authors state that they "focus on the black-box setting where defenders have no information about other training configurations (e.g., loss function and model architecture) and can only access the model via Encoder as a Service (EaaS)" and that "defenders can only retrieve feature vectors via the model API." However, Section 4.5.2 notes, "we can only use the predicted probability vectors of the input samples," which seems inconsistent. In a true black-box setting, I would expect only the predicted class ID, not output logits or probability vectors, to be available.



6. The analysis related to the amount of $D_{alt}$ in Figure 4 is essential but lacks explanation in Section 4.4.2. There is no clarification on how the authors control the ratio, whether by increasing $D_{alt}$ or reducing $D_{pub}$, or on what each point in Figure 4 represents. Since the change in log(p) for $D_{pub}$ suggests a controlled amount of $D_{pub}$, it may not be appropriate. With a fixed $D_{pub}$, only the amount of $D_{alt}$ should be adjusted.

**Questions:**

See above

---

> ### Author Response · Authors · 2024-11-21
> **Rebuttal (1/3)**
>
> Thank you for your detailed review and valuable feedback. We are grateful for the chance to respond to your comments.
>
> > **W1&W2: Task performance differences between seen and unseen data are not commonly used for verification due to two primary reasons.**
>
> We are deeply grateful for your invaluable question. Concerning the possibility that enhanced contrastive learning could eventually achieve perfect generalization of representations, we contend that, despite continuous progress in this area, models frequently fall short of such ideal generalization when confronted with complex data distributions and limited data scales. Current research [1,2,3] suggests that encoders are highly susceptible to overfitting on training data. Building on this tendency, [2] introduced membership inference techniques targeting contrastive pre-trained models, while [3] developed dataset inference methods for self-supervised encoders to mitigate model theft. In a similar vein, our work leverages this characteristic to devise a dataset ownership verification approach specifically for contrastive pre-trained models, thus addressing an important gap in this domain. Naturally, we intend to address this limitation in future studies to further enhance the generality of our method.
>
> With respect to the comment that "it is difficult to prove that the observed low $p$-values are caused by the protected data", we argue that a low $p$-value (less than 0.05) signifies a statistically significant relationship between the suspicious model’s training data and the protected data, and that merely having coincidentally similar samples is insufficient to evoke this statistical property. To substantiate this claim, we conducted extensive experiments employing various contrastive learning methods and encoder architectures on two datasets, $\mathcal{D}\_{pub}$ and $\mathcal{D}\_{alt}$, which share extremely similar distributions but no overlapping samples. The results unequivocally demonstrate that our approach can reliably distinguish models trained on $\mathcal{D}\_{pub}$ and $\mathcal{D}\_{alt}$ across all tested scenarios.
>
> ### Reference
> [1] He X, Zhang Y. Quantifying and mitigating privacy risks of contrastive learning[C]//Proceedings of the 2021 ACM SIGSAC Conference on Computer and Communications Security. 2021: 845-863.
>
> [2] Liu H, Jia J, Qu W, et al. Encodermi: Membership inference against pre-trained encoders in contrastive learning[C]//Proceedings of the 2021 ACM SIGSAC Conference on Computer and Communications Security. 2021: 2081-2095.
>
> [3] Dziedzic A, Duan H, Kaleem M A, et al. Dataset inference for self-supervised models[J]. Advances in Neural Information Processing Systems, 2022, 35: 12058-12070.

---

> ### Author Response · Authors · 2024-11-21
> **Rebuttal (2/3)**
>
> > **W3: Could the authors compare more recent related works?**
>
> Thank you for your question. In the appendix of original paper, we provide a comparison between our method and the CTRL [4] backdoor watermarking, one of the most advanced backdoor techniques designed for self-supervised models. Specifically, we injected CTRL triggers as watermarks into a small subset of data. During the verification phase, we input both watermarked and non-watermarked images into the suspicious model. If the representations of the watermarked images are significantly more similar than those of the non-watermarked images, we can conclude that the suspicious model was pre-trained on the protected dataset. The results in Appendix A.11 indicates that although methods based on CTRL can accurately identify cases where public datasets have been stolen, they also wrong the innocent suspect. For detailed results, please refer to Appendix A.11 in the attached PDF.
>
> Additionally, following your suggestion, we utilized the state-of-the-art backdoor attack method for contrastive learning [5], named CorruptEncoder, to design the watermark. Specifically, we randomly selected 10 classes from ImageNet as the public dataset $\mathcal{D}_{pub}$, referred to as ImageNet-10, which is non-overlapping with ImageNette and ImageWoof. Following the setup in the original paper, we injected CorruptEncoder's backdoor watermark into 0.5% of the images in this dataset (poisoning rate of 0.5%). The verification process is identical to that of the CTRL-based method. The results of suspicious models pre-trained on different datasets are shown below. All suspicious models are ResNet18 pre-trained using SimCLR. The shadow model is a ResNet18 pre-trained on ImageWoof using SimCLR.
>
> | Method         | ImageNet-10 | CIFAR10 | CIFAR100 | ImageNette |
> | :----          | :----       | :----   | :----    | :----      |
> | CorruptEncoder | $10^{-35}$  | 0.02    | 0.93     | $10^{-24}$ |
> | Ours           | $10^{-14}$  | 0.99    | 0.74     | 0.71       |
>
> Note that the public dataset $\mathcal{D}_{pub}$ is ImageNet-10. Therefore, the $p$-values for suspicious models trained on ImageNet-10 should be less than 0.05, while those for models trained on other datasets should be greater than 0.05. The results indicate that although the CorruptEncoder watermarking method can identify illegal behavior, it also falsely accuses innocent suspicious models. This is similar to the results obtained from experiments using CTRL.
> This occurs because backdoor watermarks (triggers) are usually fixed patterns (e.g., specific small squares for CorruptEncoder or fixed-frequency noise for CTRL), to ensure the backdoor is successfully embedded into the model. These watermarked images tend to have similar features, leading the encoder to generate more similar representations for them compared to clean images, even if the encoder is trained on a non-watermarked dataset.
>
> ### Reference
> [4] Li C, Pang R, Xi Z, et al. An embarrassingly simple backdoor attack on self-supervised learning[C]//Proceedings of the IEEE/CVF International Conference on Computer Vision. 2023: 4367-4378.
>
> [5] Zhang J, Liu H, Jia J, et al. Data Poisoning based Backdoor Attacks to Contrastive Learning[C]//Proceedings of the IEEE/CVF Conference on Computer Vision and Pattern Recognition. 2024: 24357-24366.
>
> > **W4: Section 4.5.2 is critical, as contrastive learning is often used as a pretraining method, and adversaries are more likely to release fine-tuned models.**
>
> Thank you for your advice. We also apologize for any inconvenience caused. In fact, fine-tuning the encoder for downstream tasks is one of the most common scenarios and should be given significant attention. In the experiments described in Section 4.5.2, we fine-tuned the encoder, a ResNet50 pre-trained on ImageNet using SimCLR, on CIFAR10/CIFAR100 using a learning rate of 0.001, a batch size of 512, a weight decay of 5e-4, and the SGD optimizer with a momentum of 0.9. Following your suggestions, we measured the performance of the fine-tuned model on downstream tasks and moved the experimental results into the main text. The accuracy after fine-tuning is shown in the table below.
>
> Fine-tuning on CIFAR10:
>
> | Epoch | $p$ | Acc |
> | :-- | :-- | :-- |
> | 50 | $10^{-4}$ | 0.87 |
> | 100 | $10^{-3}$ | 0.88 |
> | 150 | $10^{-8}$ | 0.88 |
> | 200 | $10^{-4}$ | 0.89 |
>
> Fine-tuning on CIFAR100:
>
> | Epoch | $p$ | Acc |
> | :-- | :-- | :-- |
> | 50 | $10^{-6}$ | 0.44 |
> | 100 | $10^{-4}$ | 0.50 |
> | 150 | $10^{-5}$ | 0.63 |
> | 200 | $10^{-3}$ | 0.66 |
>
> It indicates that our method remains effective in this more arduous scenario. For more details, please refer to Section 4.5.2 in the attached PDF.

---

> ### Author Response · Authors · 2024-11-21
> **Rebuttal (3/3)**
>
> > **W5: In a true black-box setting, only the predicted class ID, not output logits or probability vectors, to be available.**
>
> Thanks for the comment. In Section 4.5.2, we treat the fine-tuned classifier as a black-box model. Regarding the defender's capability, we refer to and follow prior dataset ownership verification methods in supervised learning [6,7], assuming that the defender can access the predicted probability vectors via the black-box model's API. As for the more challenging black-box scenario where the defender can only obtain predicted class labels, we will explore this in future work.
>
> ### Reference
> [6] Guo J, Li Y, Wang L, et al. Domain watermark: Effective and harmless dataset copyright protection is closed at hand[J]. Advances in Neural Information Processing Systems, 2024, 36.
>
> [7] Li Y, Zhu M, Yang X, et al. Black-box dataset ownership verification via backdoor watermarking[J]. IEEE Transactions on Information Forensics and Security, 2023, 18: 2318-2332.
>
> > **W6: The analysis related to the amount of $\mathcal{D}_{alt}$ in Figure 4 is essential but lacks explanation in Section 4.4.2.**
>
> We apologize for any confusion caused, and we will provide a more detailed explanation of Section 4.4.2. In Section 4.4.2, we investigated the impact of different proportions of $\mathcal{D}\_{pub}$ and $\mathcal{D}\_{alt}$ used by the suspicious model during training on our method. Specifically, we varied $r$, where $r=|\mathcal{D}\_{pub}|/|\mathcal{D}\_{pub}\cup\mathcal{D}\_{alt}|$ represents the proportion of $\mathcal{D}\_{pub}$ in $\mathcal{D}\_{pub}\cup\mathcal{D}\_{alt}$, and $\mathcal{D}\_{pub}\cup\mathcal{D}\_{alt}$ is always CIFAR10. For example, when $r=0.1$, $\mathcal{D}\_{pub}$ is 10% of the CIFAR10 training set randomly sampled, while $\mathcal{D}\_{alt}$ consists of the remaining 90%. Similarly, when $r=0.2$, $\mathcal{D}\_{pub}$ is 20% of the CIFAR10 training set randomly sampled, and $\mathcal{D}\_{alt}$ is the remaining 80%. Each point in Figure 4 represents the $p$-value (log-transformed) of the model trained on the corresponding dataset. Note that for different $r$, both $\mathcal{D}\_{pub}$ and $\mathcal{D}\_{alt}$ are different. This design aligns with the experimental setup in Section 4.2, where $\mathcal{D}\_{pub}$ is a random half of the CIFAR10 training set and $\mathcal{D}\_{alt}$ is the other half ($\mathcal{D}\_{pub}\cup\mathcal{D}\_{alt}$ is CIFAR10), effectively corresponding to $r=0.5$.
>
> The wording in the original text has been revised to be clearer based on the above meaning. For details, please refer to Section 4.4.2 of the attached PDF.
>
> In addition, based on your suggestion, we conducted experiments by fixing $\mathcal{D}\_{pub}$ and only adjusting the size of $\mathcal{D}\_{alt}$. Here, $\mathcal{D}\_{pub}$ is always 10% of the CIFAR10 training set randomly sampled, and $\mathcal{D}\_{alt}$ is randomly sampled from the remaining data. We denote $|\mathcal{D}\_{alt}|/|\mathcal{D}\_{pub}|$ as $r'$. Using ResNet18 and SimCLR for these experiments, the results are as follows. The shadow model is a ResNet18 pre-trained using SimCLR on CIFAR100.
>
> The $p$-value of the model trained on $\mathcal{D}\_{pub}$ ($p$-value should be less than 0.05):  $p=10^{-18}$
>
> The $p$-values of the models trained on $\mathcal{D}\_{pub}\cup\mathcal{D}\_{alt}$ ($p$-value should be less than 0.05):
> | $r'$ | 1 | 2 | 3 | 4 | 5 | 6 | 7 | 8 | 9 |
> | :---- | :---- | :----  | :---- | :---- | :---- | :----  | :---- | :---- | :----  |
> | $p$  | $10^{-16}$ | $10^{-15}$ | $10^{-14}$ | $10^{-13}$ | $10^{-12}$ | $10^{-12}$ | $10^{-10}$ | $10^{-10}$ | $10^{-10}$ |
>
> The $p$-values of the models trained on $\mathcal{D}\_{alt}$ ($p$-value should be greater than 0.05):
> | $r'$ | 1 | 2 | 3 | 4 | 5 | 6 | 7 | 8 | 9 |
> | :---- | :---- | :----  | :---- | :---- | :---- | :----  | :---- | :---- | :----  |
> | $p$  | $0.17$ | $0.29$ | $0.24$ | $0.39$ | $0.37$ | $0.41$ | $0.41$ | $0.57$ | $0.62$ |
>
> The results demonstrate that our method exhibits strong robustness to the size of $\mathcal{D}\_{alt}$.

---

> > ### Comment · Reviewer_F2ST · 2024-11-24
> >
> > Thank you for the authors' response.
> >
> > However, I feel that my major concerns remain unaddressed.
> >
> > This work uses the level of overfitting as a verification metric. As I previously mentioned, this approach is nearly equivalent to using high confidence scores (probability vectors) or logit values as verification metrics in categorical classification tasks.
> >
> > Confidence-based verification for classification is outdated and contrary to future advancements in training classification models. Consequently, the proposed method seems to inherit this limitation.
> >
> > In other words, both the proposed method and confidence-based verification are constrained to current strategies, which I believe is a significant limitation.
> >
> > While this work extends the confidence-based approach to the contrastive learning scenario by defining alternative metrics (two relationships) to replace confidence scores, these relationships are closely tied to the objectives of contrastive learning. This limits their novelty and reinforces the inherited limitations of confidence-based verification in classification tasks.
> >
> > Additionally, I raised a question about whether the reliability of low $p$-values arises solely from seen data, questioning the admissibility of the proposed verification.
> >
> > The authors provided empirical results using two datasets with similar distributions. However, this does not address my concern, as I specifically referred to sample-level similarity, not dataset-level distribution.
> >
> > If highly similar samples exist, low $p$-values would naturally occur. To address this, the authors should compare the Top-1 sample-level similarity between all samples in $D_{pub}$ and $D_{alt}$. Given that $D_{alt}$ is likely to be much larger than $D_{pub}$ in practice, the existence of very similar samples is highly probable.
> >
> > To resolve the admissibility issue, the authors must either empirically demonstrate that $p$-values decrease even with highly similar samples (not merely at the distribution level) or provide a theoretical justification, as the proposed method fundamentally relies on overfitting.
> >
> > Therefore, I'm sorry but I also keep my rating.

---

> ### Author Response · Authors · 2024-11-25
>
> Thank you for your feedback and for highlighting the concerns raised. We recognize that your main concerns lie in the novelty of the proposed verification metrics and the applicability of our method to datasets exhibiting high sample-level similarity. In response to these concerns, we present the following experiments and analyses:
>
> > **The distinction between our verification metric and confidence-based method.**
>
> We would like to argue that our method is **fundamentally distinct** from confidence-based method in application scenarios, methodology and performance.
>
> - **Application scenarios**. Above all, confidence-based methods are confined to supervised classification models, where confidence is well-defined. In contrast, our work centers on dataset ownership verification for contrastive pre-trained models, where neither confidence nor logits are explicitly defined. The motivation and application scenarios of two approaches are different.
>
> - **Methodologies**. Unlike confidence-based methods, our approach doesn't solely rely on the individual confidence of a single sample for verification. Instead, it calculates contrastive relationship gap by unary relationships and binary relationships, utilizing these relationships as the core verification metric. We contend that such an approach cannot be seen as a mere extension of confidence-based methods.
>
> - **Performance**. A practical comparison between our method and confidence-based methods can be made using pretrained models fine-tuned for classification tasks. As detailed in the experiments below, our method demonstrates clear superiority over confidence-based approaches.
>
> Given these differences, it is difficult to equate our approach to confidence-based methods. As a pioneering effort in dataset ownership verification for contrastive pre-trained models, we believe our contributions to this field are substantial.
>
> ### Experiment Details
> We conducted experiments on fine-tuned classification models to enable a direct comparison between two methods. Specifically, suspicious model is a classifier fine-tuned on CIFAR10, using a ResNet50 backbone pre-trained with SimCLR on ImageNet. Fine-tuning learning rate was 1e-3.
>
> Confidence-based verification is executed by comparing the highest confidence scores of samples from $D_{pub}$ training and test sets. The underlying rationale is that the higher a sample's confidence score, the more likely it belongs to the model's training set. Hence, if training set samples from $D_{pub}$ consistently exhibit higher confidence scores than those from test set, this strongly suggests that the model was trained on $D_{pub}$.
>
> Results on CIFAR10 are as follows, which show that confidence-based verification fails to detect illicit behavior, while our method detects it, further reinforcing the distinction between our method and confidence-based method in classification tasks.
>
> **Fine-Tuning on CIFAR10:**
>
> |Epoch|Acc|$p$ (Confidence-based Verification)|$p$ (Our Method)|
> |-|-|-|-|
> |50|0.87|0.23|$10^{-4}$|
> |100|0.88|0.10|$10^{-3}$|
> |150|0.88|0.15|$10^{-8}$|
> |200|0.89|0.30|$10^{-4}$|
>
> > **Whether the reliability of low $p$-value arises solely from seen data, questioning the admissibility of the proposed verification.**
>
> Through rigorous experiment, we demonstrate that our method doesn't falsely accuse a suspicious model, even when trained on a dataset with sample-level similarity. Specifically, $D_{pub}$ consists of 10,000 random images from CIFAR10 training set (20%). To simulate a dataset with sample-level similarity, we trained a ResNet50 classifier on entire CIFAR10 and used its backbone to obtain representations. For each sample in $D_{pub}$, we finded its most similar image from remaining 40,000 images in CIFAR10 training set, based on their representations' cosine similarity. These 10,000 most similar images form $D_{sim}$, which can be regarded as exhibiting sample-level similarity to $D_{pub}$. The average cosine similarity of the representations for 10,000 pairs of most similar samples was 0.97.
>
> We define the set of samples in CIFAR10 training set, excluding $D_{pub}$ and $D_{sim}$, as $\hat{D}$. Here, $D_{alt}$ is composed of $D_{sim}$ and random images from $\hat{D}$. Shadow model is a ResNet18 pre-trained by SimCLR on SVHN, while suspicious model is a ResNet18 pre-trained by SimCLR or SimSiam. Results on different $D_{alt}$ are as follows, which show that our method can perform reliably even on datasets with sample-level similarity.
>
> ||$D_{sim}$|$D_{sim}$ + 10,000 Samples from $\hat{D}$|$D_{sim}$ + 20,000 Samples from $\hat{D}$|$D_{sim}$ + 30,000 Samples from $\hat{D}$|
> |-|-|-|-|-|
> |SimCLR|0.74|0.39|0.25|0.70|
> |SimSiam|0.88|0.96|0.81|0.91|
>
> We once again express our gratitude for your insightful comments. We trust that these clarifications address your concerns, and we would be most appreciative if you could kindly reassess our work with careful consideration.

---

> ### Comment · Reviewer_F2ST · 2024-11-26
>
> 1. I believe my main concern has not been accurately conveyed. While I understand the stated differences (e.g., target tasks, verification metrics), the performance comparison remains inadequate since the targeted scenarios differ, as the authors mentioned. My point was that both approaches (confidence-based and the proposed method) rely on evaluation metrics specific to the learned tasks (supervised classification and contrastive learning).
> For the proposed method, as I understand it, contrastive learning aims to produce well-clustered representations for variants of a single image and well-separated representations for distinct images. SimCLR achieves this using pairwise similarity as its objective function. In supervised classification, accurate predictions with higher confidence are achieved by minimizing cross-entropy.
>
> In both cases, verification depends on task-specific metrics under the assumption that better metrics on seen data result from overfitting. However, this dependency negatively aligns with model generality, which is expected to improve in the future. I am concerned about the misalignment between the proposed verification approach and the direct advancements in contrastive learning.
>
> If I extend this issue to cases like LLMs or image segmentation, the proposed method essentially translates into a verification approach based on producing results that are "more similar" to the ground truth. For instance, in the case of LLMs, if the model generates an answer that is more similar to the ground truth for a input text included in the training set, I question whether this should be considered data cheating purely based on the similarity of the result. Similarly, for image segmentation, it is also questionable to suspect a model reporting high performance on seen data as cheating. This seems to align with the same concept underlying the proposed method.
>
>
> Contrastive learning also has their ground truth, which is automatically generated. Basically, it directly minimizes the two relationships defined in the proposed method, indicating directly correlated with the objective of contrastive learning and the verification metrics. Given the binary decision nature of contrastive learning, the proposed method uses two datasets: one that can be "cheated" and another that cannot.
>
> This dichotomy highlights a fundamental similarity in the verification mechanism across these scenarios. Ultimately, these cases raise the same question: is achieving better results on seen data inherently indicative of data cheating, or does it simply reflect the overfitting properties of the model? This distinction is crucial to evaluate the general applicability and robustness of the proposed method.
>
>
> 2. Regarding the reliability issue, I apologize for my earlier mistake. I mistakenly referred to $D_{pvt}$ as $D_{pub}$ when discussing scenarios where highly similar images to $D_{pvt}$ exist in $D_{alt}$. Since $D_{alt}$ is typically much larger than $D_{pvt}$, such scenarios seem plausible. I now ask whether the proposed method is robust when $D_{alt}$ includes highly similar images to $D_{pvt}$.
> While additional empirical evaluations may not be feasible due to time constraints, a theoretical discussion on this matter would help clarify the method’s robustness.

---

> > ### Comment · Reviewer_F2ST · 2024-11-26
> >
> > However, I think the current empirical results show another kind of robustness and partially address the issue, so I’ll consider it. Please answer focusing on the first issue, negative alignment to advancement of contrastive learning .

---

> ### Author Response · Authors · 2024-11-27
>
> We sincerely appreciate your prompt and insightful feedback. We are pleased to have partially resolved your concerns and are honored to engage in continued dialogue with the reviewer. In this response, we aim to address the remaining issues by reflecting upon the following question.
>
> > **Will the contrastive relationship gap be entirely eliminated with the advancement of contrastive learning?**
>
> This is a thought-provoking and open-ended question, one that invites a diversity of perspectives. While we concur that, with the continued evolution of contrastive learning, the contrastive relationship gap may diminish, we hold that it will not be wholly eradicated. If such an eventuality were to occur, it would signal the near culmination of AI development, as overfitting represents one of the most fundamental challenges in the field.
>
> Even if, in a distant future (say, ten years hence), the contrastive relationship gap were to be fully overcome, we firmly believe that the current work—pioneering in the realm of dataset ownership verification for contrastive models—remains of considerable significance. It serves as a catalyst for the exploration of more sophisticated approaches in the development of contrastive learning. At the very least, it offers a robust and effective solution for the current landscape of contrastive learning methodologies.
>
> > **When $D_{pvt}$ and $D_{alt}$ have highly similar samples.**
>
> To illustrate that our method does not unjustly classify legitimate suspicious models, even when $D_{alt}$ contains samples highly similar to those in $D_{pvt}$, we conducted the following supplementary experiments.
>
> Similarly, $D_{pub}$ consists of 10,000 randomly selected images from the CIFAR10 training set (20%), with $D_{pvt}$ representing the CIFAR10 test set. To simulate a dataset exhibiting sample-level similarity to $D_{pvt}$, we trained a ResNet50 classifier on the entire CIFAR10 dataset and utilized its backbone to extract feature representations. For each sample in $D_{pvt}$, we identified the most similar image from the remaining 40,000 images in the CIFAR10 training set (excluding $D_{pub}$), based on the cosine similarity of their representations. These 10,000 most similar images form $D_{sim}$, which can be considered as exhibiting a sample-level similarity to $D_{pvt}$. The average cosine similarity between the representations of the 10,000 most similar sample pairs was 0.98.
>
> We define the set of samples in CIFAR10 training set, excluding $D_{pub}$ and $D_{sim}$, as $\hat{D}$. Here, $D_{alt}$ is composed of $D_{sim}$ and random images from $\hat{D}$. Shadow model is a ResNet18 pre-trained by SimCLR on SVHN, while suspicious model is a ResNet18 pre-trained by SimCLR or SimSiam.
>
> In this scenario, the suspicious model remains legitimate, which implies that the $p$-value should exceed 0.05. The results obtained with varying $D_{alt}$ are shown below, demonstrating that even when $D_{pvt}$ and $D_{alt}$ exhibit sample-level similarity, our approach does not mistakenly flag legitimate suspicious models.
>
> ||$D_{sim}$|$D_{sim}$ + 10,000 Samples from $\hat{D}$|$D_{sim}$ + 20,000 Samples from $\hat{D}$|$D_{sim}$ + 30,000 Samples from $\hat{D}$|
> |-|-|-|-|-|
> |SimCLR|0.77|0.37|0.62|0.70|
> |SimSiam|0.99|0.82|0.39|0.91|

---

> ### Author Response · Authors · 2024-12-01
>
> Dear reviewer `F2ST`, as the discussion period draws to a close, we kindly ask if our response has sufficiently addressed your concerns. We are more than willing to provide further clarification on any remaining issues. Thank you for reviewing our paper.

---

> > ### Comment · Reviewer_F2ST · 2024-12-01
> >
> > I sincerely appreciate the authors' thoughtful responses.
> >
> > I agree that:
> >
> > 1) The problem of data copyright in contrastive learning is significant.
> > 2) The proposed manuscript demonstrates strong performance across multiple evaluations.
> >
> > Additionally, all other reviewers have already rated this work highly, resulting in a high average score.
> >
> > I understand that addressing my final concern, which pertains to advancements in contrastive learning, is challenging and may need to remain an open question. However, my concern lies in the limited usefulness and admissibility in terms of verification. Furthermore, I think the proposed method represents a direct adaptation of confidence-based verification to contrastive learning, which raises questions about its novelty. As a reviewer, I think this aspect should be informed to AC prior to the AC's recommendation process.
> >
> > Except for this concern, all my other issues have been empirically addressed, thanks to the authors' responses.

---

> ### Author Response · Authors · 2024-12-02
>
> We deeply appreciate your meticulous efforts in reviewing this work. We are pleased to have addressed nearly all of your concerns. Regarding the final issue, we have offered clear clarifications distinguishing our approach from confidence-based methods in prior reply. While we respectfully disagree with your viewpoint, we fully understand your perspective as a reviewer on this matter. Once again, we thank you for your invaluable contributions to the review of this work.

---

### Official Review · Reviewer_RYsp · 2024-11-02

**Soundness:** 3
**Presentation:** 3
**Contribution:** 2
**Rating:** 8
**Confidence:** 3

**Summary:**

The paper proposes a novel method for dataset ownership verification (DOV) specifically tailored for contrastive pre-trained models in self-supervised learning. The method utilizes two observations about contrastive learning: the unary and binary relationships in the embedding space of models trained on a specific dataset. These observations are exploited through a contrastive relationship gap metric, calculated between the suspected model and a shadow model pre-trained without the dataset in question. Comprehensive experiments across datasets and models demonstrate that the method effectively detects unauthorized dataset usage, outperforming baseline techniques.

**Strengths:**

1. **Innovative Approach:** The method uniquely applies to self-supervised models by leveraging characteristics of contrastive learning, filling a gap in current DOV methods that primarily target supervised learning.
2. **Black-box Applicability:** The approach is suitable for black-box scenarios, which is practical and aligned with real-world applications where full model access is unavailable. The approach demonstrates robust performance across different datasets (e.g., CIFAR, ImageNet) and architectures, indicating generalizability.
3. **Effective Performance:** Results show high sensitivity, specificity, and AUROC scores, suggesting that the proposed metric reliably distinguishes between legitimate and unauthorized dataset use.
4. **Efficiency:** Compared to alternatives, the method is computationally efficient, which enhances its applicability to large datasets like ImageNet.
5. **Thorough evaluation:** The paper is very comprehensive, they make specific claims and justify them with solid experiments and results.

**Weaknesses:**

1. **Dependency on Feature Representation Access:** The method requires access to feature representations, which might not be feasible in all practical scenarios, as many services limit this access for security reasons.

2. **Limited Application to Non-Contrastive Pre-Trained Models:** The method’s effectiveness is constrained to contrastive learning. Other prevalent pre-training strategies, such as masked image modeling (MIM), are not effectively addressed, potentially limiting applicability. However, the authors make a clear claim and explain this as a limitation, thus I think the weaknesses are ok given this is one of the earlier works.

**Questions:**

1. Could early stopping or other techniques be leveraged to reduce the effectiveness of this detection method? A brief section on possible attack approaches in the appendix may be considered.

The paper is quite extensive very thorough! :)

---

> ### Author Response · Authors · 2024-11-21
> **Rebuttal**
>
> Thank you for your thorough review and constructive feedback. We appreciate the opportunity to address your concerns.
>
> > **W1: The method requires access to feature representations, which might not be feasible in all practical scenarios.**
>
> Thanks for the insightful comment. Here the proposed Dataset Ownership Verification (DOV) method is specifically tailored for self-supervised pre-trained models, designed as a universal feature extractor rather than being tailored to specific downstream tasks.  Moreover, we focus on the black-box setting where defenders lack access to training configurations (e.g., loss function and model architecture), and can only retrieve feature vectors via its model API. This assumption aligns with the current landscape where self-supervised models are accessible via Encoder as a Service (EaaS) [1,2,3].  The issue of Dataset Ownership Verification for encoders has not been previously explored.  In this context, we introduce our method to authenticate the data sources of black-box encoders, bridging this security void and promoting a more secure EaaS environment.
>
> ### Reference
> [1] Dziedzic A, Duan H, Kaleem M A, et al. Dataset inference for self-supervised models[J]. Advances in Neural Information Processing Systems, 2022, 35: 12058-12070.
>
> [2] Liu Y, Jia J, Liu H, et al. Stolenencoder: stealing pre-trained encoders in self-supervised learning[C]//Proceedings of the 2022 ACM SIGSAC Conference on Computer and Communications Security. 2022: 2115-2128.
>
> [3] Sha Z, He X, Yu N, et al. Can't steal? Cont-steal! Contrastive stealing attacks against image encoders[C]//Proceedings of the IEEE/CVF Conference on Computer Vision and Pattern Recognition. 2023: 16373-16383.
>
> > **W2: The proposed method's applicability is limited to contrastive learning models.**
>
> Thank you for your feedback. We agree that broadening the scope of the method to encompass more self-supervised techniques could amplify its utility and influence. This paper, serving as a pioneering endeavor towards this direction, focuses on dataset ownership verification within contrastive pre-trained models, one of the most representative and well-developed self-supervised approaches. We plan to tackle the issue of DOV for more self-supervised pre-trained models in our future research endeavors.
>
> > **Q1: Could early stopping or other techniques be leveraged to reduce the effectiveness of this detection method?**
>
> Thanks for raising the question. The early stopping technique can terminate model training prematurely, which may result in less pronounced contrastive relationship gap. To investigate the impact of early stopping on our method, we specifically set the patience of early stopping (the maximum number of epochs allowed to continue training when the K-Nearest Neighbors accuracy on the validation set does not improve significantly over multiple consecutive epochs) to 15 and 30, respectively. We then calculated the $p$-values of the trained models using the same method, as shown in below table. Both the datasets of defender and suspect are CIFAR10, meaning $p$-value should be less than 0.05. Self-supervised method is SimCLR. The shadow model is a ResNet18 pre-trained on CIFAR100 using SimCLR. The results demonstrate that our method remains effective even under early stopping conditions.
>
> As per your suggestion, we have added this section to Appendix A.10. For details, please refer to the attached PDF.
>
> | Model | w/o Early Stopping | w/ Early Stopping (patience=15) | w/ Early Stopping (patience=30) |
> | :---- | :---- | :----  | :---- |
> | ResNet18  | $10^{-12}$ | $0.01$ | $10^{-4}$ |
> | VGG16  | $10^{-11}$ | $10^{-4}$ | $10^{-5}$ |

---

> > ### Comment · Reviewer_RYsp · 2024-11-23
> >
> > Thank you for your response, I have read through the rebuttal and the updated paper. I will keep my score.

---

> ### Author Response · Authors · 2024-11-23
>
> We are grateful for your timely suggestions! They have significantly contributed to the refinement of our work.

---

### Official Review · Reviewer_Vzgv · 2024-11-03

**Soundness:** 3
**Presentation:** 3
**Contribution:** 2
**Rating:** 8
**Confidence:** 4

**Summary:**

This paper introduces a novel method for verifying dataset ownership in self-supervised contrastive learning models, protecting curated datasets from unauthorized use. The proposed approach leverages the "contrastive relationship gap," capturing distinct similarities in representations for models trained on a defender's dataset versus unrelated data. Through a three-step process, the method identifies unauthorized usage efficiently and with high accuracy. Experimental results on several datasets and models show that this approach outperforms existing methods in accuracy, computational efficiency, and robustness, even under privacy-preserving settings like DP-SGD, making it a promising tool for dataset security in machine learning.

**Strengths:**

- The paper presents a unique dataset ownership verification (DOV) method specifically tailored for self-supervised contrastive learning models. This is a valuable addition to the field, as existing DOV methods are generally focused on supervised or non-contrastive learning models, leaving a gap that this paper addresses.
-The authors conduct extensive experiments across multiple datasets (CIFAR10, CIFAR100, SVHN, ImageNet variants) and contrastive learning architectures (SimCLR, BYOL, MoCo, DINO), demonstrating the method's effectiveness and generalizability. This robust experimental setup strengthens the validity of the proposed approach.
- By requiring only a small subset of the defender’s data for verification, the proposed method is more computationally efficient than baseline methods like D4SSL, which require access to the entire dataset, making it suitable for large-scale applications.

**Weaknesses:**

- While the proposed method offers a novel approach to dataset ownership verification, its applicability is limited to contrastive learning models. Many self-supervised learning models use objectives other than contrastive learning, so expanding the method’s scope could enhance its impact. However, this limitation is relatively minor.
- In line 488, the authors state that "the private training method does not affect our verification results," but this claim is based on experiments using only DP-SGD with a high privacy budget (epsilon=50). To support this claim, it would be beneficial to test the method under stronger privacy settings or with alternative privacy-preserving techniques, such as differentially private generative models.

**Questions:**

- How effective is the approach when the suspect model undergoes adaptation, such as fine-tuning on a different dataset or with altered weights? Is the method still robust under such conditions, or are there specific scenarios where fine-tuning could mask the original dataset’s influence on the model?
- How sensitive is the proposed DOV method to variations in the suspect model’s hyperparameters or pre-training dataset characteristics (e.g., domain-specific data)?

---

> ### Author Response · Authors · 2024-11-21
> **Rebuttal**
>
> Thank you for your thorough review and constructive feedback. We appreciate the opportunity to address your concerns.
>
> > **W1: The proposed method's applicability is limited to contrastive learning models.**
>
> Thank you for the nice suggestion. We agree with the reviewer that broadening the scope of the method to encompass more self-supervised techniques could amplify its utility and influence. This paper, serving as a pioneering endeavor towards this direction, focuses on dataset ownership verification within contrastive pre-trained models, one of the most representative and well-developed self-supervised approaches. We plan to tackle the issue of DOV for more self-supervised pre-trained models in our future research endeavors.
>
> > **W2: It would be beneficial to test the method under stronger privacy settings.**
>
> Thank you for your advice. Following your suggestion, we have conducted experiments using stronger privacy settings. Specifically, we reduced the epsilon parameter in DP-SGD to observe its impact on our method. The shadow model is a ResNet18 pre-trained using SimCLR on SVHN. Both the public dataset $\mathcal{D}_{pub}$ and the suspicious dataset are ImageNette. The suspicious model is a ResNet18 pre-trained using SimCLR. The results below demonstrate that our method exhibits good performance within a wide range of the privacy budget.
>
> | $ \epsilon $ | 1         | 5         | 10         | 20         | 30         | 40         |
> | :----   | :----      | :----      | :----      | :----      | :----      | :----      |
> | $p$     | $10^{-9}$ | $10^{-10}$ | $10^{-10}$ | $10^{-12}$ | $10^{-13}$ | $10^{-13}$ |
>
> > **Q1: How effective is the approach when the suspect model fine-tunes on a different dataset?**
>
> Thanks for the comment. To evaluate the proposed method with finetuned models, we used a ResNet18 pre-trained on ImageNet with SimCLR and fine-tuned it on CIFAR100 using different contrastive learning methods. The fine-tuning parameters include a learning rate of 1e-3, a weight decay of 5e-4, and a batch size of 512. The shadow model is a ResNet18 pre-trained using SimCLR on SVHN. Both the public dataset $\mathcal{D}_{pub}$ and the suspicious dataset are ImageNet. The results for different fine-tuning epochs are as follows, which indicates that our method remains effective in this more arduous scenario.
>
> | Fine-tuning Epoch | 0          | 10        | 20        |
> | :----             | :----      | :----     | :----     |
> | SimCLR            | $10^{-3}$ | $10^{-3}$ | $0.01$ |
> | SimSiam           | $10^{-4}$  | $10^{-4}$ | $10^{-3}$ |
>
> > **Q2: How sensitive is the proposed method to variations in the suspect model’s hyperparameters or pre-training dataset characteristics?**
>
> Thank you for your question. We analyzed whether suspicious model's hyperparameter would affect our method. Specifically, we set different batch size (64 for the shadow model and 32 for the suspicious model), learning rate (0.06 for the shadow model and 0.01 for the suspicious model), and weight decay (5e-4 for the shadow model and 1e-4 for the suspicious model) for the shadow model compared to the suspicious model. The shadow model is a ResNet18 pre-trained using SimCLR on SVHN. Both the public dataset $\mathcal{D}_{pub}$ and the suspicious dataset are ImageNette. The suspicious model also employs a ResNet18 architecture. Our method exhibits commendable resilience to the training hyperparameter settings of the suspicious model.
>
> | Self-supervised Method | All Same | Different Batch Size | Different Learning Rate  | Different Weight Decay |
> | :---- | :---- | :----  | :---- | :---- |
> | SimCLR  | $10^{-11}$ | $10^{-4}$ | $10^{-4}$ | 0.02 |
> | BYOL | $10^{-10}$ | $10^{-3}$ | $10^{-3}$ | $10^{-4}$ |
> | SimSiam | $10^{-5}$  | $10^{-4}$ | $10^{-3}$ | $10^{-5}$ |
>
> Furthermore, we analyzed the impact of the pre-training dataset of the suspicious model on our method. We used a ResNet18 pre-trained on SVHN with SimCLR as the shadow model and tested it on suspicious models pre-trained on different datasets. The suspicious model is a ResNet18 pre-trained using SimCLR. In each case, the public dataset was the training set for the suspicious model, meaning $p$-values are expected to be less than 0.05. The results, as shown below, indicate that our method demonstrates good robustness to the pre-training dataset of the suspicious model.
>
> | Suspect's Dataset | CIFAR10 | CIFAR100 | ImageNette | ImageWoof |
> | :---- | :---- | :----  | :---- | :---- |
> | $p$  | $10^{-11}$ | $10^{-14}$ | $10^{-8}$ | $10^{-16}$ |

---

> > ### Comment · Reviewer_Vzgv · 2024-11-26
> > **Reply**
> >
> > I appreciate the authors' additional efforts in the rebuttal. They have thoroughly addressed all my concerns, and I am satisfied with their responses and will raise my score to accept.

---

> ### Author Response · Authors · 2024-11-27
>
> We appreciate your response and the increased score! Your suggestions have greatly helped us refine and improve our work.

---

### Author Response · Authors · 2024-11-21

We thank all reviewers for their helpful feedback and will address raised concerns individually under each review. We are glad the reviewers find that:

* Our paper addresses a critical security issue: dataset ownership verification in contrastive learning.

    * "This is a valuable addition to the field, as existing DOV methods are generally focused on supervised or non-contrastive learning models, leaving a gap that this paper addresses." - `Vzgv`

    * "The method uniquely applies to self-supervised models by leveraging characteristics of contrastive learning, filling a gap in current DOV methods that primarily target supervised learning." - `RYsp`

    * "This paper addresses an important and novel problem—dataset copyright protection in contrastive learning." - `F2ST`

    * "The paper introduces a novel method for dataset ownership verification (DOV) specifically tailored for contrastive pre-trained models, addressing a critical need in data rights protection." - `KUSb`

    * "The research topic is important." - `DVkE`

* Our experiments are well-conducted.

    * "The authors conduct extensive experiments across multiple datasets and contrastive learning architectures , demonstrating the method's effectiveness and generalizability." - `Vzgv`

    * "The paper is very comprehensive, they make specific claims and justify them with solid experiments and results." - `RYsp`

    * "The method has been validated across multiple contrastive pre-trained models, demonstrating its broad applicability." - `KUSb`

    * "The authors conduct many experiments." - `DVkE`

* Our method achieves outstanding performance.

    * "Through a three-step process, the method identifies unauthorized usage efficiently and with high accuracy." - `Vzgv`

    * "Results show high sensitivity, specificity, and AUROC scores, suggesting that the proposed metric reliably distinguishes between legitimate and unauthorized dataset use." - `RYsp`

    * "The authors provide a comprehensive range of experiments, and the proposed method consistently demonstrates outstanding results across all tested settings." - `F2ST`

    * "Experimental results show that the method can significantly outperform previous methodologies with a high probability of rejecting the null hypothesis (p-value well below 0.05)." - `KUSb`

    * "The performance is strong compared with baselines." - `DVkE`

* Our method is highly efficient.

    * By requiring only a small subset of the defender's data for verification, the proposed method is computationally efficient, making it suitable for large-scale applications. - `Vzgv`

    * Compared to alternatives, the method is computationally efficient, which enhances its applicability to large datasets like ImageNet. - `RYsp`


In response to the feedback, we have made the following modifications to our submission (the revised sections are highlighted in blue in the attached PDF):

- Added an appendix section to present visualization results.
- Added an appendix section to present the impact of early stopping on our method.
- Moved some important results from the appendix to the main text.
- Adjusted the size of certain images to comply with the page limit.
- Corrected the issues in phrasing as pointed out by the reviewers.

We hope that our responses and revisions to the paper can increase the reviewers' confidence in our work. We would be extremely grateful if this could lead to a higher score, especially from reviewers `F2ST` and `DVkE`. :)

---

### Meta-Review · Area_Chair_GKjH · 2024-12-17

**Metareview:**

The reviewers largely vote for acceptance.  There is one reviewer that assigned the paper a 5, but I think their one remaining point of contention is not a strong reason for rejection.  I recommend acceptance.  Having read through the weaknesses pointed out by reviewers, the authors have addressed these alleged weaknesses extensively, including revisions to the draft to address concerns about the writing and also including many new experiments such as applying their method to CLIP which I think can make their paper more broad.  There are two fundamentally unresolved points.  (1) The method is only applicable to contrastive learning.  While it is true that there are many competitors to contrastive learning, I think this limitation is fine since there is a lot of work on contrastive learning alone and that should not be disqualifying here.  (2) As phrased by one reviewer, “Enhanced contrastive learning might eventually generalize representations, clustering representations from single-sample into a single point.”  I find this claim to be dubious, and I think this method will remain applicable in the long term.  Overall, both I and the reviewers are very positive about this paper.

**Additional Comments On Reviewer Discussion:**

The authors responded heavily to the reviews and have largely addressed all substantial feedback, including paper revisions.

---

### Decision · Program_Chairs · 2025-01-22

Accept (Poster)